# "Pour some sugar on me"—Environmental *Candida albicans* isolates and the evolution of increased pathogenicity and antifungal resistance through sugar adaptation

Theresa Lange[1], Jakob L. Sprague[1], Raghav Vij[1], Raquel Alonso-Roman[1,2], Nadja Jablonowski[1], Silvia Radosa[3,4], Thomas Krüger[5], Olaf Kniemeyer[5], Falk Hillmann[3,6], Axel A. Brakhage[5,7], Stefanie Allert[1], Sascha Brunke[1]*, Bernhard Hube [1,2,7]*

1 Department of Microbial Pathogenicity Mechanisms, Leibniz Institute for Natural Product Research and Infection Biology, Hans Knoell Institute, Jena, Germany, 2 Cluster of Excellence Balance of the Microverse, Jena, Germany, 3 Evolution of Microbial Interactions, Leibniz Institute for Natural Product Research and Infection Biology, Hans Knoell Institute, Jena, Germany, 4 ASKION GmbH, Gera, Germany, 5 Department of Molecular and Applied Microbiology, Leibniz Institute for Natural Product Research and Infection Biology, Hans Knoell Institute, Jena, Germany, 6 Biochemistry/Biotechnology, Faculty of Engineering, Wismar University of Applied Sciences Technology, Business and Design, Insel Poel, Germany, 7 Institute of Microbiology, Friedrich Schiller University, Jena, Germany

* bernhard.hube@leibniz-hki.de (BH); sascha.brunke@leibniz-hki.de (SB)

## Abstract

*Candida albicans* is an opportunistic fungal pathogen that colonizes mucosal surfaces of most humans. Only in rare cases, *C. albicans* isolates are found in the environment. This study investigated whether environmental isolates differ in their virulence potential from clinical strains and how adaptation to a human diet influences key virulence attributes. We examined three *C. albicans* isolates from oak trees in the United Kingdom, and observed that one exhibited high host cell damage, increased hypha formation, invasion capacity, and candidalysin production, along with an intrinsic resistance to amphotericin B. The other two showed lower virulence which was still similar to most tested clinical isolates. All oak tree isolates showed an increased resistance to fluconazole. To mimic the more recent evolution of *C. albicans* to a sugar-rich diet, we evolved a low-damaging isolate in sugar-rich medium, which unexpectedly enhanced its metabolic flexibility, epithelial damage potential, and antifungal resistances, including a new resistance to amphotericin B. These findings suggest that *C. albicans* isolates can develop high virulence potential and antifungal resistance in the environment, and that adaptation of *C. albicans* to sugar-rich diets, as in westernized countries, can affect fungal pathogenicity and drug resistance.

**Data availability statement:** The whole genome sequencing data (PRJEB88226) and transcriptome data (PRJEB88454) generated in this study are uploaded in the European Nucleotide Archive database.

**Funding:** T.L., S.B., and B.H. received funding from the Priority Program SPP2225 "Exit strategies of intracellular pathogens" (project number 446404928) of the German Research Foundation (Deutsche Forschungsgemeinschaft – DFG). J.L.S., T.K., O.K., A.A.B, and B.H. received support from the DFG within the Collaborative Research Centre (CRC)/Transregio 124 FungiNet Project (project ID 210879364; projects A1, C1, and Z2). F.H. was supported by German Research Foundation/Deutsche Forschungsgemeinschaft (DFG), Grant no. HI 1574/4-1. R.V. and B.H. were supported by the ANR/BMBF 2019 Antimicrobial resistance call, project titled "Antifungal 323 Resistance: From Surveillance to Treatment" (AReST). R.A.R. and B.H. were funded by the Deutsche Forschungsgemeinschaft (DFG, German Research Foundation) under Germany´s Excellence Strategy - EXC2051 - Project ID 390713860. S.A. and B.H. were supported by the DFG project Hu 528/20-1. The funders had no role in study design, data collection and analysis, decision to publish, or preparation of the manuscript.

**Competing interests:** The authors have declared that no competing interests exist.

## Author summary

In this study, we set out to understand whether *Candida albicans* found in nature can be as dangerous as the strains collected from patients, and whether a diet high in sugar affects the fungal pathogenicity. We began by comparing three fungal isolates taken from oak trees in the United Kingdom to strains isolated in hospitals. To our surprise, one tree-derived strain showed stronger virulence attributes than most clinical strains – it invaded human cells more readily, produced significant amounts of a toxin that damages host cells, and survived treatment with antifungals.

To explore how modern, sugar-rich diets might influence virulence traits of *C. albicans*, we "trained" a less harmful tree isolate by growing it in a sugar-rich medium over a longer period of time. This process made it more flexible in how it uses nutrients, more aggressive toward human cells, and less susceptible to several antifungal treatments.

Our findings demonstrate that *C. albicans* populations outside of direct human contact can acquire high pathogenic potential and drug resistance. Moreover, our study shows that exposure to sugar-rich diets may boost *C. albicans* ability to infect humans and evade treatments.

## Introduction

The yeast *Candida albicans* is closely associated with humans as a commensal and an opportunistic pathogen, an association that can potentially be traced back to the evolution of early hominids [1]. During commensalism, *C. albicans* can be found in the oral and vaginal cavity as well as in the gastrointestinal tract, where it is normally kept in check by immune surveillance mechanisms and the microbiota [2–4]. When the host immune system or intestinal barriers are compromised, or when the microbiota balance is disturbed, the fungus can turn pathogenic [3]. During pathogenicity, *C. albicans* produces hyphae, invades and damages tissues. In severe cases, the fungus can translocate into the bloodstream. This can lead to dissemination throughout the body and infection of vital organs [3,5]. Both as a commensal and as a pathogen, *C. albicans* constantly must adapt to its in-host niches. This includes coping with physiological temperature, adherence to host surfaces and formation of biofilms, competition with other microbes, acquisition of nutrients, and immune evasion [3,6,7].

Few environmental *C. albicans* isolates have been described so far and it has been suggested that they may rather be contaminations [8], as the majority of them were found in areas influenced by humans – such as hospital surroundings [9,10], urban aquatic [11–13] and soil environments in cities [14,15], as well as on fruit and vegetable sources [16]. However, more recently, *C. albicans* strains were isolated from non-urban environmental niches such as soil, trees and other plants, and pigeon droppings

[17–22]. In 2016, three *C. albicans* strains were found in the New Forest national park, UK on ancient oak trees [17,20], indicating that these yeasts may not only be transiently or accidentally associated with trees. All three isolates were found to be phylogenetically distinct [17], and interestingly, one oak tree strain is phylogenetically different from any known clade [17]. This suggests an environmental reservoir of yet undescribed *C. albicans* clades.

Contrary to the predictions from an earlier study, which found a *C. albicans* ancestor associated with early hominids [1], it has been hypothesized that *C. albicans* may not be an evolutionary old commensal of the human host, but that it has been acquired more recently due to the influence of therapeutic treatments (especially antibiotics) and the adoption of western diets [23]. Supporting this, humans living in remote communities have a low prevalence of only 3% for gastrointestinal *C. albicans* [24]. In contrast, in westernized countries, *C. albicans* can be found as a commensal in approximately 30–90% of healthy individuals [25–28]. Diets enriched in carbohydrates, which are hallmarks of western-style diets, have been shown to facilitate growth and gut colonization of *C. albicans in vivo* [29–31]. Hence, the question arises whether sugar-rich diets consumed in western countries could have supported fungal adaptation, to colonize, and to consequently infect the human host.

In this study, we asked whether phenotypes of *C. albicans* environmental isolates differ from clinical isolates in terms of virulence attributes, host cell damage, and intrinsic antifungal resistance. By characterizing the above-mentioned oak tree isolates [17,20], we identified a highly virulent, antifungal-resistant *C. albicans* strain from the environment. In contrast, the other two environmental isolates showed a lower virulence capacity similar to clinical isolates. To mimic fungal adaptations to sugar-rich diets and its consequences to virulence and antifungal resistance, we used one of the low-damaging environmental *C. albicans* isolates for a laboratory evolution experiment. Long-term exposure to a host dietary sugar selected for an increased virulence potential and metabolic plasticity in this isolate. Furthermore, the sugar-adapted strain evolved antifungal resistances. These data show that environmental isolates can have an unexpectedly high level of virulence and antifungal resistance and that sugar-rich diets can affect both fungal pathogenicity and antifungal susceptibility.

## Results

### Phenotypic differences among the environmental *Candida albicans* isolates

We used the three oak trees isolates (S1 Table: Strains used in this study) to characterize their phenotypes and investigate the possibility of an environmental reservoir for pathogenic *C. albicans* strains. To this end, we investigated their growth under different conditions and survival upon amoebae predation, and then tested clinically relevant parameters, including virulence attributes and antifungal susceptibility.

First, we evaluated the metabolic plasticity of the oak tree isolates by assessing their growth on a synthetic defined medium supplemented with different carbon (Fig 1A) and nitrogen (Fig 1B) sources. Importantly, these included compounds that are present in the human gut, such as the host dietary sugars galactose, fructose, and sucrose [32,33], and alternative carbon and nitrogen sources like short-chain fatty acids, *N*-acetylglucosamine (GlcNAc), and amino acids [34–36]. Significant differences in the patterns of nutrient use were found across the environmental strains. One of the *C. albicans* oak tree isolates, Oak 2, grew comparable to the reference strain SC5314 in most tested carbon sources, with the exception of citric acid, acetate, and citrate (Fig 1A). Oak 3 grew similarly well in most tested carbon sources, except for a few sugars such as galactose and maltose, and showed significantly better growth in citric acid. Oak 1 showed reduced growth in host dietary sugars like glucose, fructose, and galactose, as well as in multiple alternative sugars such as maltose, fucose, rhamnose, raffinose, and ribose. Similar to Oak 3, Oak 1 grew better in citric acid than SC5314. Moreover, Oak 1 showed deficits in growth on all tested alternative nitrogen sources, except for arginine (Fig 1A, B). Oak 3 grew slower on certain nitrogen sources like phenylalanine, tryptophan, and with low levels of ammonium sulfate. Finally, Oak 2 grew similarly to SC5314 on the tested nitrogen sources (Fig 1B), with the exception of phenylalanine, tryptophan, and very low concentrations of ammonium sulfate. In general, the oak tree isolates were not unusual in their nutrient utilization spectrum, as we observed similarly reduced growth (compared to SC5315) in galactose and maltose for at least two of the four tested clinical *C. albicans* isolates (S1A Fig). Moreover, all four clinical isolates had a somewhat lower growth in alternative nitrogen sources, including

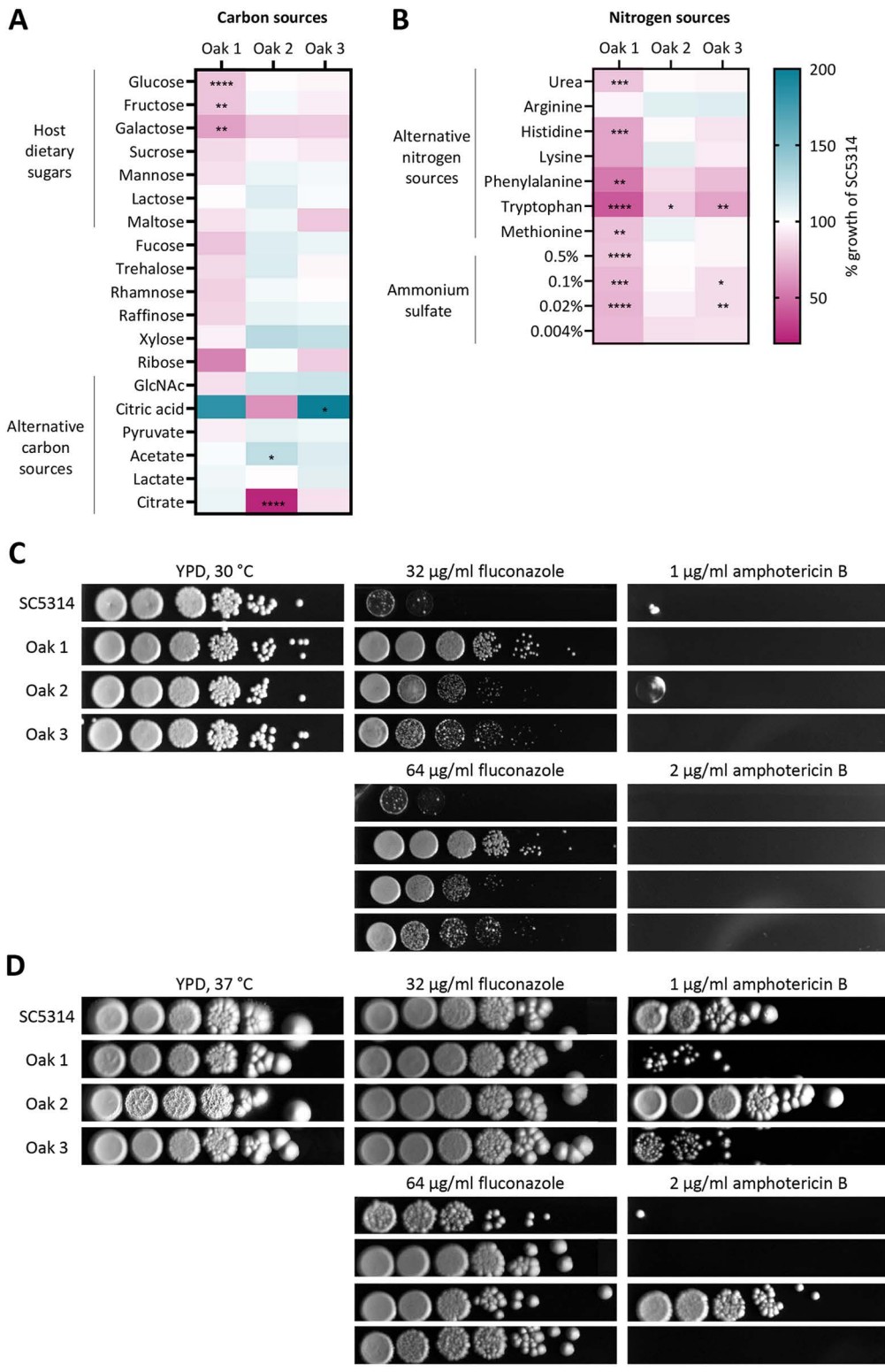

**Fig 1. The oak tree strains display significant differences in nutrient use, and Oak 2 shows resistance to amphotericin B.** Growth of all three oak tree isolates was assessed as growth curves at 30 °C in different media comprising specific carbon (A) or nitrogen (B) sources. The growth is shown as area under the curve relative to the reference strain SC5314 (% growth). Asterisks indicate significance compared to SC5314 calculated using a one-way ANOVA with Tukey's multiple comparisons test (* p < 0.05, ** p < 0.01, *** p < 0.001, **** p < 0.0001) (n = 3). (C, D) Resistance to fluconazole and

amphotericin B was determined by performing a drop test of serially diluted *C. albicans* solutions on YPD agar plates supplemented with the indicated antifungal concentration. Pictures were taken (C) after 2 days at 30 °C and (D) after 6 days at 37 °C. For all pictures, contrast and brightness were increased by 20%.

amino acids such as phenylalanine, tryptophan, histidine, and lysine (S1B Fig), similar to Oak 1 and Oak 3. Overall, the nutrient utilization spectra of clinical and environmental isolates therefore overlapped significantly for our tested strains.

We hypothesized that the oak tree isolates were potentially exposed to amoeboid predators in the environment, similar to other environmental fungal pathogens [37,38], which may have selected for the ability to withstand such attacks. To test this, we evaluated the survival of the oak tree strains upon amoeba predation after 3 h. For all three environmental isolates, almost 100% of the inoculum survived the amoebae contact, which is significantly more than SC5314 (54% inoculum survival) (S1C Fig). Moreover, when analyzing the phagocytosis rate of the oak tree isolates after 2 h of amoeba predation, Oak 1, Oak 2 and Oak 3 were significantly less phagocytosed by the amoebae than SC5314 (S1D Fig), indicating a potential adaptation to amoeba predation.

We assumed that the oak tree isolates had not been in contact with clinically prescribed antimycotics. We therefore tested their antifungal drug susceptibilities by performing drop tests of serial dilutions on agar plates supplemented with commonly prescribed antifungals, namely azoles and polyenes. All oak tree isolates, especially Oak 1, showed better growth in the presence of fluconazole than SC5314 (Fig 1C). Most surprisingly, when the strains were tested for growth in the presence of amphotericin B (AmB), Oak 2 showed a high degree of resistance (Fig 1D).

### One oak tree *C. albicans* isolate with reference strain-like host cell damage potential

We expected that the environmental strains are not adapted to mammals, and therefore tested the damaging potential of all three oak tree isolates on epithelial cells. Host cell damage was determined *via* lactate dehydrogenase (LDH) release 24 h and 48 h after infection of oral or intestinal epithelial cells, respectively, to account for the different infection kinetics with the different host cells (Fig 2A, B). We found that Oak 1, Oak 3, and the tested clinical isolates did not significantly damage the oral and intestinal epithelial cells, when compared to SC5314 (Fig 2A, B). Interestingly, the AmB-resistant strain, Oak 2, induced damage to intestinal epithelial cells at a level similar to the high-damaging SC5314 strain [39,40] (55% of complete lysis), and reached 64% of the full lysis control on oral epithelial cells (Fig 2A, B). On both epithelial cell types, Oak 2 was among the most damaging strains tested.

To investigate the basis for this damaging potential further, its underlying features – hypha formation, adhesion, and invasion – were quantified on oral epithelial cells. Adhesion of all oak tree isolates to host cells was comparable to SC5314 after 1 h, except for Oak 3, which displayed a 2-fold higher, albeit not statistically significant, level of adhesion (Fig 2C). Oak 1 and Oak 3 showed a significantly lower rate of filamentation (26% and 42%, respectively) and invasion compared to SC5314 on oral cells (Fig 2D, E), and grew mainly as yeast cells. Interestingly, Oak 2 showed SC5314-like filamentation (99%) and invasion into oral epithelial cells.

To further characterize the virulence potential of Oak 2, the survival rate of this strain was quantified as compared to SC5314 in an *ex vivo* human whole blood model [41]. Oak 1 was included as a non-virulent, environmental control strain. Remarkably, both strains, the high-damaging Oak 2 and the low-damaging Oak 1, survived significantly better in whole human blood than SC5314 over 4 h, with a 3-fold higher number of surviving cells at the final time point (Fig 2F). Collectively, Oak 2 could therefore represent a new *C. albicans* clade with high damaging potential and a high level of antimicrobial resistance, which can also survive and thrive outside of the human host.

### Differences in candidalysin sequence and expression kinetics among the oak-derived isolates

We went on to investigate the basis for the high damage potential of Oak 2. The most important contributor to host cell damage by *C. albicans* is candidalysin (CaL), a hypha-associated pore-forming peptide toxin encoded (together with other

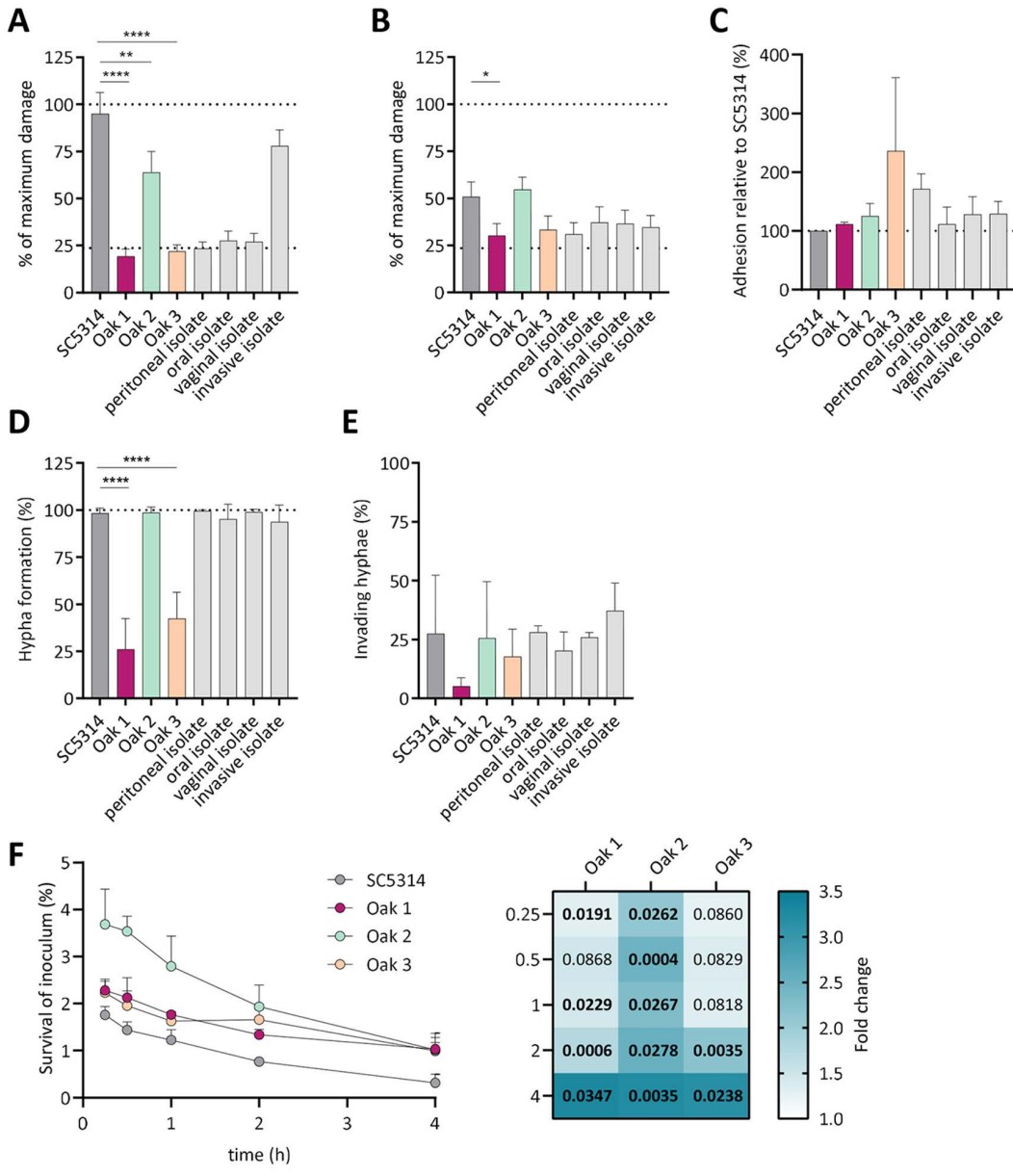

**Fig 2. One oak tree *C. albicans* isolate shows an increased virulence potential on epithelial cells and augmented survival in an *ex vivo* blood model.** (A) Damage to oral epithelial cells by the environmental oak tree strains and clinical isolates was determined after 24 h by LDH release (n = 4). The lower dashed line indicates the uninfected control. The upper dashed line indicates the full lysis control. One-way ANOVA with Tukey's multiple comparisons test (** $p < 0.01$, **** $p < 0.0001$). (B) Damage to intestinal cells was monitored 48 h post infection by quantifying LDH release (n = 3). One-way ANOVA with Tukey's multiple comparisons test (* $p < 0.05$). (C) Adhesion of environmental strains was determined relative to SC5314 at 1 h post-infection of oral cells by staining and microscopy (n = 3). (D) Hypha formation on oral cells was determined by microscopic evaluation 3 h post infection. One-way ANOVA with Tukey's multiple comparisons test (**** $p < 0.0001$) (for oak strains and SC5314: n = 3-6, clinical isolates: n = 2-3). (E) Invasion was measured by differential staining of oral cells 3 h post infection. Invasion was calculated as percentage of invading hyphae compared to all counted hyphae (n = 3). (F) Survival of *C. albicans* in whole human blood was assessed over the course of 4 h by plating surviving colony-forming units (4 donors). Survival is shown relative to the inoculum. Heat map displays fold change to SC5314 and p-values calculated with a two-way ANOVA with repeated measures and Dunnett's multiple comparison test. Significant p-values are indicated in bold.

peptides) by the *ECE1* gene [42–44]. We therefore determined the transcript levels of *ECE1* during infection of epithelial cells by quantitative real-time PCR (Fig 3A, B). On oral epithelial cells, Oak 2 expressed lower levels of *ECE1* transcripts compared to SC5314 (fold change of 0.5) at both tested time points (24 h and 48 h post infection), but the levels were still significantly higher than in Oak 1 and Oak 3 (Fig 3A). In fact, these strains did not produce measurable transcript levels of *ECE1* during *in vitro* oral infection, although the gene is present in their genomes (Fig 3C). On intestinal cells, the *ECE1* transcript level of Oak 2 reached 0.7 times that of SC5314 after 24 h, and even exceeded its level after 48 h (at 1.8-fold) (Fig 3B). Similar to oral cells, Oak 3 and Oak 1 had lower *ECE1* levels than SC5314 on intestinal cells. Only Oak 1 reached a measurable transcript level, at 0.36 times that of SC5314 (Fig 3B).

Although the amino acid sequence of Ece1 is largely conserved among *C. albicans* strains, some regions of candida-lysin and the non-candidalysin-Ece1 peptide (NCEP) sequences are known to differ between clinical isolates, which can have an impact on candidalysin-mediated host cell damage [44–46]. We, therefore, performed whole genome sequencing of the environmental strains and aligned the predicted amino acid sequences of CaL and NCEP peptides of Oak 1, Oak 2, and Oak 3 to SC5314 (Fig 3C). There is little variation in Ece1 sequences overall (Ece1 sequence similarity to SC5314: 0.971 for Oak 1, 0.989 for Oak 2, 0.982 for Oak 3), and the majority of amino acid substitutions were found in peptide 2, peptide 3 (CaL), and peptide 7. In Oak 1 and Oak 2, we identified new amino acid sequence variants of CaL (S2E Fig) that differed from the 10 known variants from 182 clinical *C. albicans* isolates [46] – somewhat supporting their

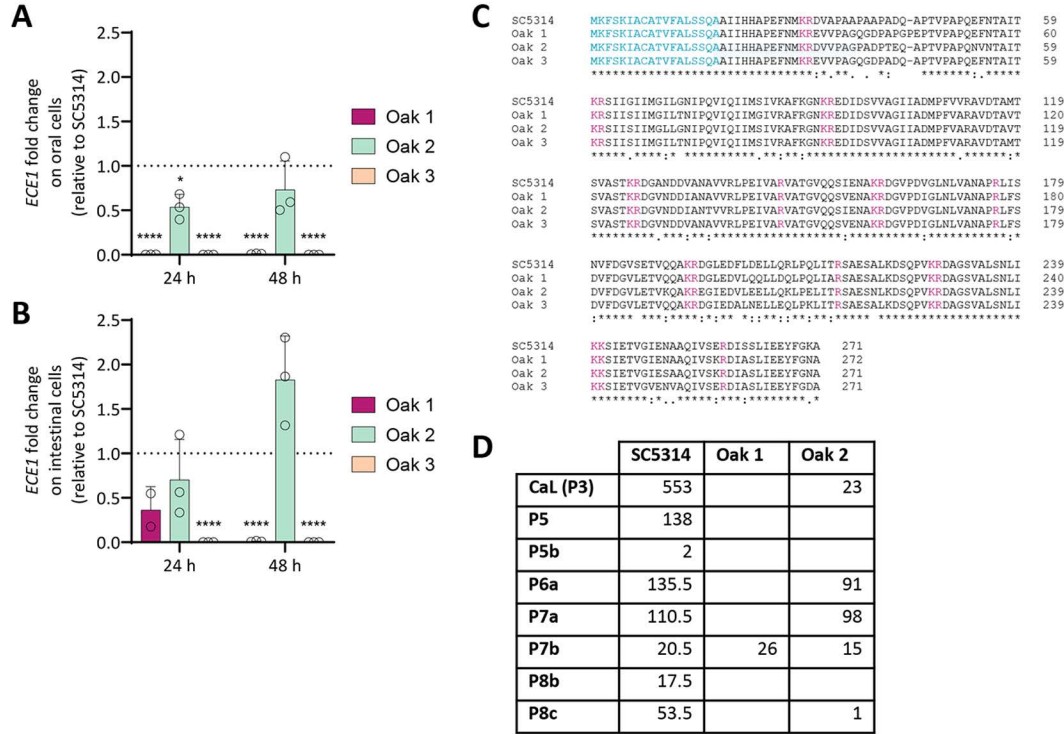

| | SC5314 | Oak 1 | Oak 2 |
|---|---|---|---|
| CaL (P3) | 553 | | 23 |
| P5 | 138 | | |
| P5b | 2 | | |
| P6a | 135.5 | | 91 |
| P7a | 110.5 | | 98 |
| P7b | 20.5 | 26 | 15 |
| P8b | 17.5 | | |
| P8c | 53.5 | | 1 |

**Fig 3. *ECE1* (Extent of cell elongation 1) characteristics of Oak 2 resemble the SC5314 reference strain.** *ECE1* expression on oral (A) and intestinal epithelial cells (B) is shown as fold change relative to SC5314 at the indicated time points. The expression is normalized to *ACT1* mRNA levels. Statistical significance was calculated using a one-way ANOVA with Tukey's multiple comparisons test (* $p < 0.05$, **** $p < 0.0001$). (C) Predicted Ece1 sequences were aligned using Clustal Omega. Identical amino acids are indicated as asterisks (*) and similar ones with dots (. or:). Blue indicates the signal sequence and pink amino acids (KR, KK, R) mark the Kex protease cleavage sites of the polypeptide. (D) Secretion of Ece1 peptides and peptide fragments from the indicated strains into culture medium, evaluated by LC-MS/MS analysis. The mean peptide-spectrum matches (PSM) scores for each identified peptide are presented (n = 2); if no number is given, the peptide was not detected. Oak 3 was not included in this measurement.

non-human origin. Only the amino acid sequence of Oak 3 is identical to one of the most common candidalysin variants identified in clinical isolates (S2E Fig) [46]. CaL sequence differences might contribute to the different levels of epithelial damage (Fig 2), as shown previously [46]. The other peptide sequences as well as the seven lysine-arginine (KR) motifs, which serve as cleavage sites for Golgi-located Kex proteases [42], are highly conserved among the environmental strains (Fig 3C). Alignments of the *ECE1* promoter sequences (S2A, B, D Fig) revealed that both, Oak 1 and Oak 3 lack one out of two TATA boxes present in the SC5314 *ECE1* promoter. While the first TATA box 109 bp upstream of the open reading frame of *ECE1* is the primary regulator of gene expression, the second box at -284 bp is considered mostly dispensable for the transcription of *ECE1* [47]. These differences may account for the lower *ECE1* transcript levels in the two oak isolates. The promoter sequence of Oak 2 contained a single TATA box at a position much further upstream (-165 bp instead of -109 bp) (S2B Fig), potentially explaining the different expression dynamics on intestinal cells, with high transcript levels at a very late time point.

Like SC5314, all oak tree isolates also code for a secretion signal sequence at the 5' end of the open reading frame of the *ECE1* gene. We investigated the secretion efficiency of the Ece1 peptides, including candidalysin, NCEPs, and the corresponding fragments [48] as another potential explanation for the differences in host cell damage. To this end, the cell-damaging Oak 2, SC5314 as a positive control, and Oak 1 as a non-damaging isolate were cultured in strong hypha-inducing conditions and the protein content of the supernatant was analyzed by LC-MS/MS [49]. As expected, we found most Ece1 peptides and peptide fragments, including candidalysin, in the SC5314 supernatant, but no peptides other than peptide 7b in the non-virulent Oak 1 (Fig 3D). Oak 2 showed a moderate secretion of candidalysin and measurable levels of peptides 6a, 7a, and 7b (Fig 3D).

Overall, our results indicate that Oak 2, but not Oak 1 or Oak 3, is capable of inducing SC5314-like damage to oral and intestinal epithelial cells due to strong filamentation and subsequent invasion into epithelial cells, associated with a strong expression of *ECE1* and secretion of candidalysin into the invasion pocket [49].

## Oak 2, but not Oak 1 or Oak 3, inflict damage to macrophages

Since Oak 2 showed a high virulence potential toward human epithelial cells and a high survival rate in whole blood, we asked whether Oak 2 would also damage immune cells and thus investigated the interaction of the three oak tree isolates with primary human macrophages. In macrophages, Oak 2 again showed SC5314-like damage after 24 h, whereas Oak 1 and Oak 3 induced significantly less damage compared SC5314, similar to the other tested clinical isolates (Fig 4A). Similarly, Oak 2 induced macrophage membrane rupture to an extent similar to SC5314 (S8B Fig). However, in this assay Oak 3 reached a level comparable to Oak 2. Oak 1 induced significantly less host cell lysis than SC5314 (S8B Fig). Interestingly, all oak tree isolates survived inside the phagocytes significantly better than SC5314 (Figs 4B and S8D), with Oak 1 having the highest survival rate (388% of SC5314) (Fig 4B). To determine the contribution of filamentous *vs.* yeast growth to these differences, we determined the fungal morphology microscopically at 3 h post infection. For Oak 1 we observed stretched macrophages filled with yeasts, reflecting its high intracellular survival ability (Figs 4C and 8E). In line with the high damage induction, Oak 2 formed filaments similar to SC5314, and Oak 3 replicated as yeasts, but less than Oak 1. In conclusion, Oak 2 exerts strong pathogenicity toward macrophages, while Oak 1 showed an increased adaptation and survival inside macrophages. To investigate whether these different phenotypes are driven by the macrophages or due to the fungal-intrinsic strain variations, we investigated the immune activation of macrophages by the oak tree isolates (S9B Fig). Interestingly, all three strains did not differ significantly in triggering IL-8, TNFα, IL-6, and IL-1β release, except for Oak 3, which induced a significantly higher level of IL-8.

## Mimicking *C. albicans*' evolution in sugar-rich medium

Sugar-rich diets consumed in westernized countries typically contain low-digestible oligosaccharides [33,50]. These are comprised of sugars such as galactose and fructose, and can reach the large intestine where *C. albicans* mainly resides as a

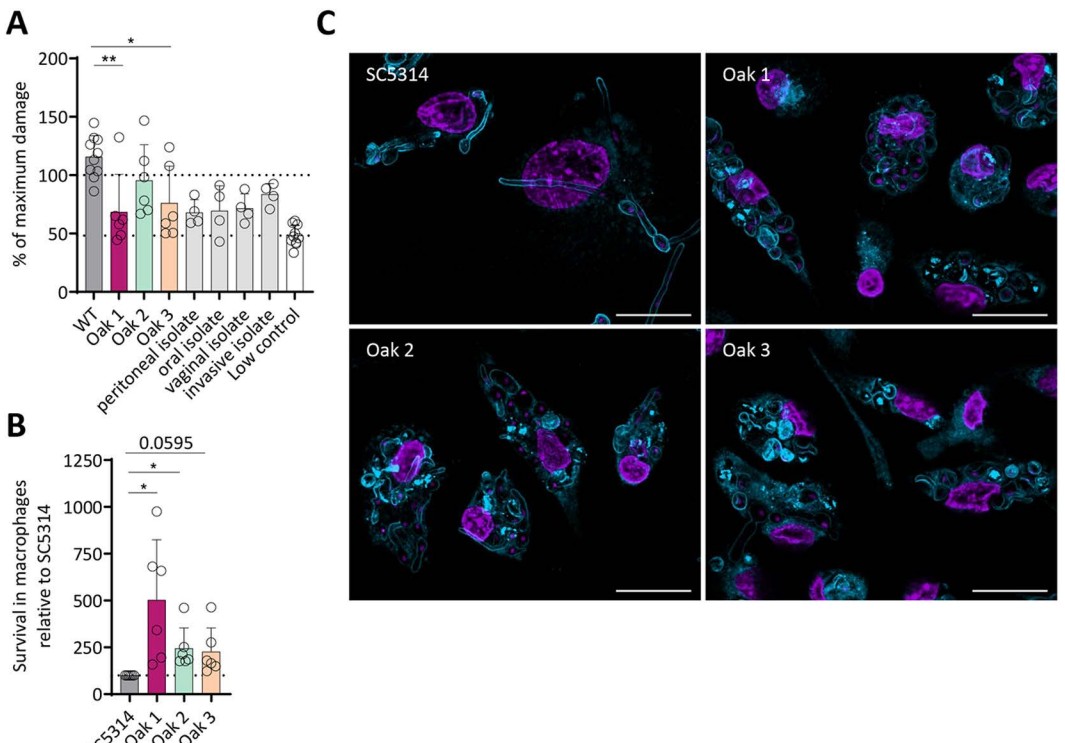

**Fig 4. The oak tree strains show differences in their interaction with primary macrophages compared to the reference strain.** (A) Damage to monocyte-derived macrophages by the environmental oak tree strains was determined after 24 h by measuring LDH release. The lower dashed line indicates the uninfected control. The upper dashed line indicates the full lysis control. Each circle represents one donor. Significance was calculated by a one-way ANOVA with Tukey's multiple comparisons test (* $p < 0.05$, ** $p < 0.01$) (n = 4-6 donors). (B) Intracellular survival of the *C. albicans* strains in human monocyte-derived macrophages was assessed 3 h post infection by lysis and plating of the intracellular fungal cells. Survival is depicted relative to the reference strain SC5314. Each circle represents one donor. Significance was calculated by a one-sample *t*-test against 100% (* $p < 0.05$) (n = 6 donors). (C) Microscopic pictures were taken either 3 h after infecting human monocyte-derived macrophages (37 °C, $CO_2$) with a 63 × magnification with immersion oil (scale bar, 20 µm). Samples were stained with DAPI (nuclei staining, purple) and Concanavalin A-AlexaFluor647 (fungal staining, blue). Figure shows representative pictures from at least 4 donors.

commensal [32,33]. Oak 1 has shown diminished growth with all tested host dietary sugars and a low virulence potential, which feasibly could be increased if the strain were to adapt better to the human host. We hypothesized that adaptations to components of westernized diets can influence *C. albicans*-host interactions, and therefore performed an *in vitro* evolution experiment that mimicked aspects of this diet change. To this end, we continuously passaged Oak 1 in triplicate with a host dietary sugar, galactose, at 2% as its sole carbon source (Fig 5A). For 6 weeks, the strain was transferred into fresh medium every two to three days, whenever it reached the late exponential growth phase. In preliminary screens for hypha formation, antifungal resistance, and interaction with host cells, two of the three evolved strains showed no significant differences compared to the parent strain (S3 Fig), and were hence not used for follow-up analyses. In contrast, one of the evolved replicates, from here on indicated as 'Evo', adapted well to growth on galactose and was therefore chosen for further investigation.

## The sugar-adapted *C. albicans* strain shows an increased metabolic flexibility and augmented resistance to antifungals

To investigate the sugar-evolved Evo strain, we re-characterized its growth on dietary sugars and alternative carbon sources which can be present in the human body [32–36]. Additionally, several different nitrogen sources were investigated.

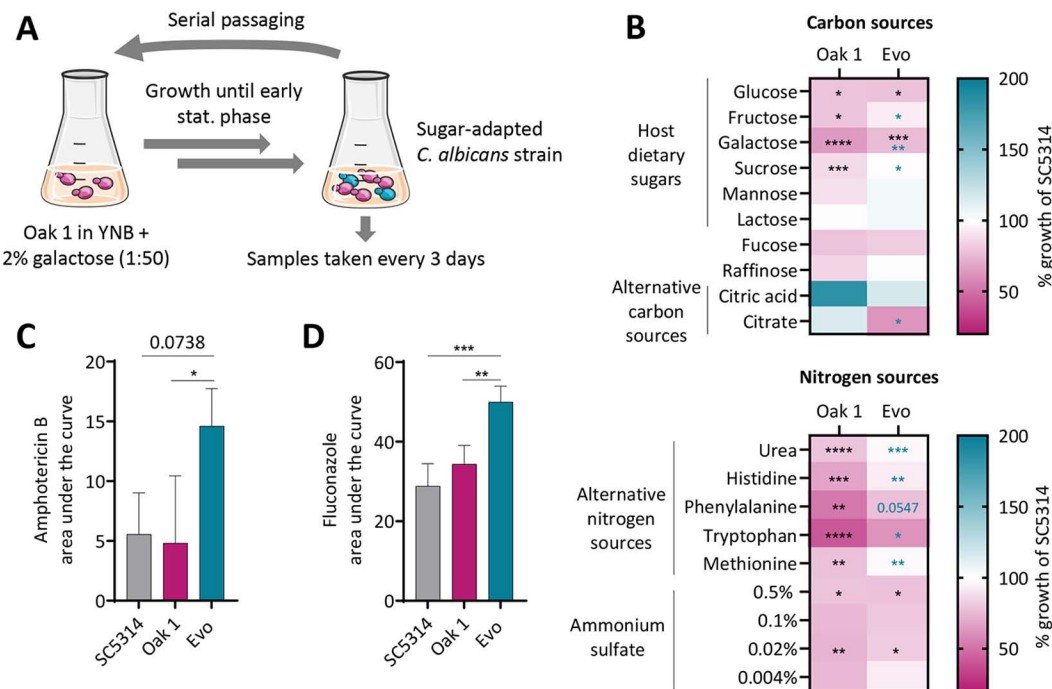

**Fig 5. Experimental evolution with a host dietary sugar leads to antifungal resistance and metabolic flexibility.** (A) Schematic overview of the evolution experiment in YNB medium supplemented with 2% galactose. Oak 1 and SC5314 overnight cultures were diluted 1:50 into the galactose-containing medium and were incubated for 6 weeks at 30 °C, with passaging them into fresh medium every 2-3 days when the early stationary phase was reached. Growth was measured as $OD_{600nm}$ over the whole time course. Samples were taken and frozen after every passage. (B) After the evolution experiment, growth of the sugar-adapted strain (Evo) was characterized in multiple carbon and nitrogen sources. The growth is shown as area under the curve relative to the reference strain SC5314 (% growth). Black asterisks indicate significance compared to SC5314 and blue asterisks indicate whether the sugar-adapted strain grew significantly different compared to its parental Oak 1 strain. Significance was calculated by using a one-way ANOVA with Tukey's multiple comparisons test (* $p < 0.05$, ** $p < 0.01$, *** $p < 0.001$, **** $p < 0.0001$) (n = 3). (C, D) Antifungal resistance of the galactose-evolved strain was assessed by performing growth curves in YPD supplemented with either 32 µg/ml fluconazole or 0.5 µg/ml amphotericin B at 30 °C for 72 h, and is depicted as area under the curve. Significance was calculated by using one-way ANOVA with Tukey's multiple comparisons test (* $p < 0.05$, ** $p < 0.01$, *** $p < 0.001$) (n = 3).

Interestingly, the sugar adaptation significantly improved growth of the evolved strain on the tested dietary sugars, including fructose, galactose, and sucrose, as well as within most of the alternative nitrogen sources, except for two higher concentrations of ammonium sulfate, compared to the parent strain (Fig 5B, indicated with blue asterisks). Thus, the long-term adaptation to galactose increased the nutritional flexibility of the evolved strain from its parental stage. However, compared to SC5314, the sugar-evolved strain still grew at a significantly lower rate in glucose, galactose, and with low levels of ammonium sulfate (Fig 5B, indicated with black asterisks). Nevertheless, the overall nutrient utilization pattern became more similar to SC5314 – for instance with citric acid, where the Evo strain grew worse than its parent, but at a rate similar to SC5314 (Fig 5B).

We then tested the sugar-adapted Evo strain for its susceptibility to the antifungals fluconazole and amphotericin B. Strikingly, Evo displayed significantly better growth and thus an increased antifungal resistance compared to its parental strain or SC5314 in the presence of either antimycotic (Fig 5C, D). It seems that long-term exposure to a host dietary sugar induced resistance to multiple classes of antifungals in this strain.

## Sugar adaptation led to a transcriptome shift in *C. albicans* during interaction with epithelial cells

In order to identify the underlying mechanism for the increased metabolic flexibility and antifungal resistances, we investigated the changes in the sugar-adapted *C. albicans* strain on a genomic and transcriptomic level. To this end, we

performed whole genome sequencing of the sugar-adapted strain and aligned it to the haploid genome of the parental Oak 1 [17]. We reliably identified nine mutations, however, we did not find a likely candidate in the form of, e.g., a change in the coding sequence of a relevant gene. The majority of mutations were single nucleotide polymorphism or frame shifts in intergenic regions, except one missense mutation in a gene coding for a transmembrane sorting receptor for vacuolar hydrolases (S3 Table). These mutations may have contributed to the observed phenotypes, but their individual effects are difficult to ascertain especially in a non-reference strain of *C. albicans*. Similarly, we did not find any likely causative mutations in the other evolved strains (S3 Table).

We therefore performed RNA sequencing of Oak 1 and 2 and the sugar-adapted Evo after 24 h and 48 h on intestinal epithelial cells, which simulate a relevant host environment. Principal component analysis (PCA) analysis revealed that the transcript profiles of both oak tree isolates cluster together, independent of the time point of infection (Fig 6A). After the sugar adaptation, a shift along the PC1 axis was observable, suggesting that the evolution on the host dietary sugar led to the transcriptional shift.

We performed functional GO term enrichment analysis to find pathways differently regulated in the environmental and sugar-adapted strains compared to SC5314 during intestinal cell infection (Fig 6B). For Oak 1, we observed a higher expression of glyoxylate cycle and amino acid metabolism genes in comparison to SC5314. Furthermore, Oak 1 showed reduced transcript levels of sugar metabolism pathways and other metabolic processes like glycolysis, autophagy, fatty acid metabolism, general carbon metabolism, and protein degradation compared to SC5314. This likely led to the diminished growth of Oak 1 on most of the tested carbon and nitrogen sources (Fig 1A, B). Most of the differentially regulated GO terms enriched in Oak 2 were also enriched in Oak 1. The sugar-adapted Evo strain showed an enrichment of highly expressed galactose, sucrose, and fructose metabolism pathway genes (Figs 6B and S4), as well as of glycolysis and autophagy, compared to its parental strain.

Based on these findings, we specifically looked at the expression of genes involved in sensing of extracellular glucose and galactose levels (*HGT4*), transport of glucose (e.g., *HGT17*, *HGT2*, etc.) and galactose (*HGT7*), as well as genes involved in signaling downstream of the sugar sensor (S4 Fig). For all these genes we observed increased transcript levels in the sugar-adapted strain compared to its parental oak tree isolate. Other host dietary sugar transporter genes like *MAL31*, and downstream signaling genes such as *MIG1*, *GAL7*, *GAL10*, and *GAL1* were similarly more transcribed in the Evo strain. The notable exception was the galactose transporter gene *HGT7*, which exhibited low transcript levels – very likely due to selection on high levels of galactose. Overall, the transcriptional changes indicate that the long-term adaptation to one host dietary sugar led to the emergence of a *C. albicans* strain adapted for better growth on multiple sugars.

Interestingly, ABC transporter gene transcript levels were higher in the sugar-evolved Evo strain than in its parental strain or SC5314. The ABC transporter class includes multidrug transporters that confer resistances to azoles [51], and their upregulation may explain the fluconazole resistance observed in the sugar-adapted strain.

### The sugar-adapted *C. albicans* strain has increased virulence potential *in vitro* and in an intestinal organ-on-chip model of infection

Next, we characterized damage induction by the sugar-evolved Evo strain on oral and intestinal epithelial cells *via* release of LDH at 24 h and 48 h post infection. On oral cells, damage by the Evo strain, like its parent, remained only slightly above the non-infected control (Fig 7A). Interestingly, on intestinal cells, we observed a drastic increase in damage that reached SC5314-like levels (Fig 7B). Thus, our laboratory evolution on dietary sugars led to an increased damage toward epithelial cells of *C. albicans*' intestinal niche.

To determine the causes for this surprising phenotype, we investigated hypha formation and invasion at 6 h after infecting intestinal epithelial cells. All three oak tree isolates as well as the sugar-adapted Evo showed shorter hyphae and less invasion compared to SC5314, while their filamentation rate was similar (S5 Fig). To investigate the phenotype of the sugar-evolved strain in the oral niche, we tested adhesion, hypha formation, and invasion with oral cells at 1 h and

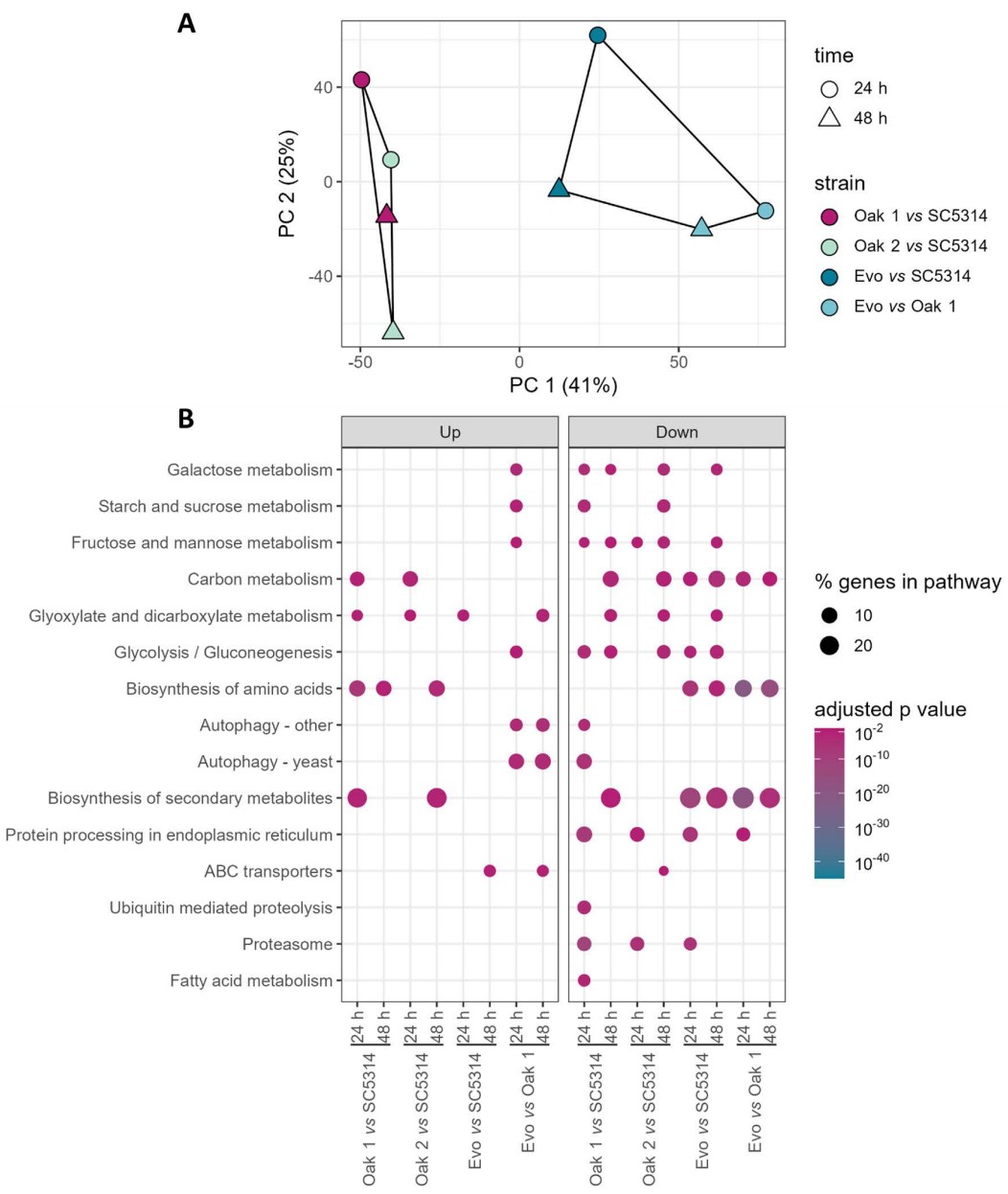

**Fig 6. The sugar adaptation leads to transcriptomic remodeling.** (A) Principal component analysis of the transcriptomes for the SC5314, two oak tree isolates, and Evo on intestinal cells. Samples clustered together depending on their origin, and the sugar-adapted strain underwent a positive shift along the PC1 axis. (B) KEGG pathways enriched for genes that were significantly up- or down-regulated on intestinal epithelial cells 24 or 48 h post infection. % genes in pathway is indicated by the dot size and represents the number of genes within each pathway that were differentially regulated compared to the indicated reference strains (below the graph). The color scale indicates the adjusted p value.

3 h post infection. Adhesion of Evo was higher compared to both, SC5314 and the parental oak tree isolate (about 270% of SC5314 and 245% of the parent adhesion level, respectively) (Fig 7C). Similarly, hypha formation and invasion of Evo significantly increased, up to SC5314-like levels (Fig 7D, F). The hyphal length, however, remained significantly shorter than SC5314 (Fig 7E).

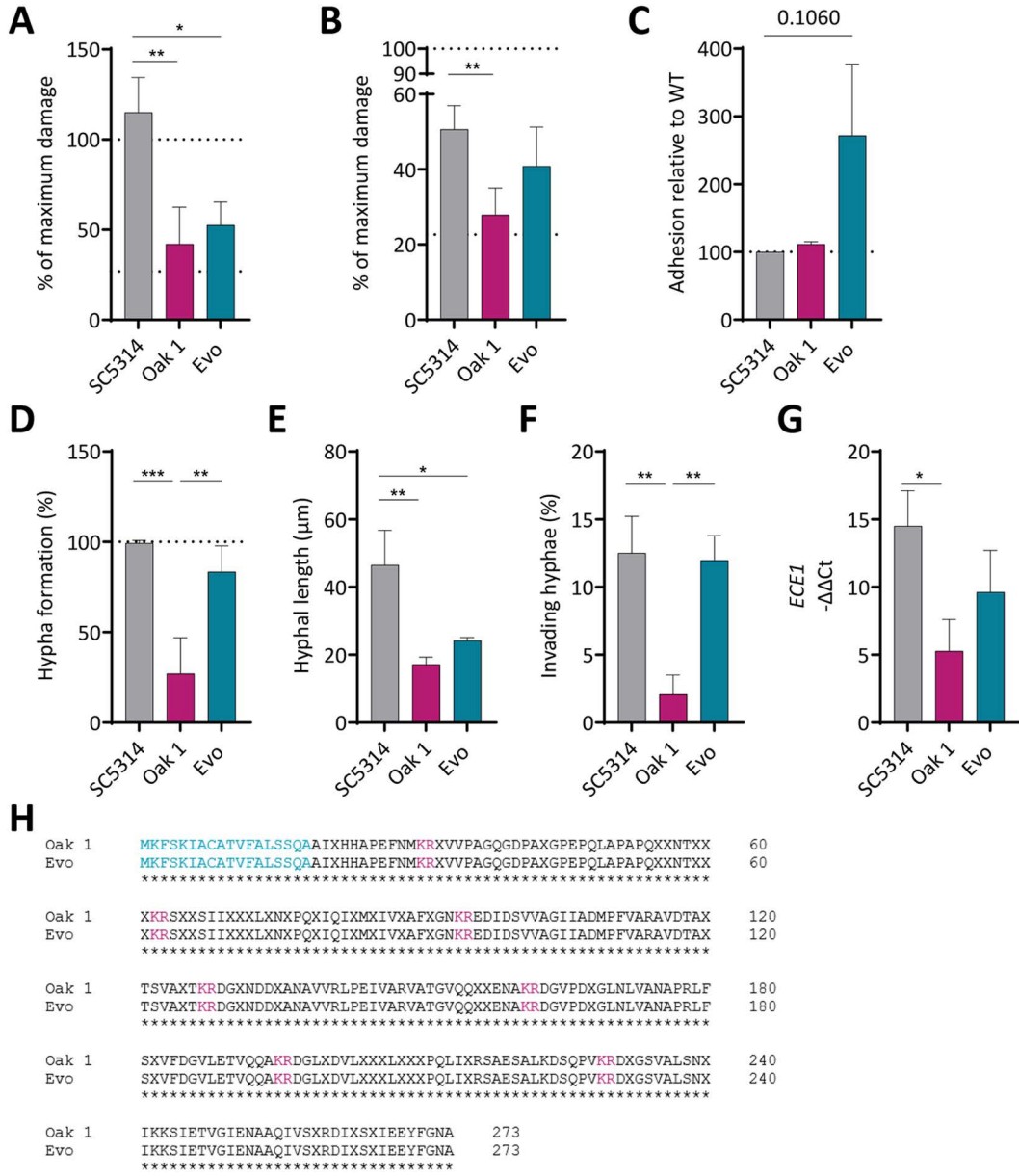

**Fig 7. The sugar-evolved Evo strain shows increased adhesion, filamentation, invasion, and host cell damage during interaction with epithelial cells.** (A) Damage to oral cells and (B) intestinal cells was evaluated by quantifying epithelial LDH release 48 h post infection (oral n = 3, intestinal n = 4). The lower dashed line indicates the uninfected control. The difference between Oak 1 and Evo was not statistically significant in (A) and (B). (C) Adhesion of *C. albicans* strains was assessed by fixing and staining the fungal cells 1 h post infection of oral cells. The average number of attached *C. albicans* cells per microscopic image is plotted relative to the reference strain SC5314 (n = 3). (D) Hypha formation was determined by microscopic evaluation counting the hypha-forming cells relative to all *C. albicans* cells on oral cells 3 h post infection (n = 3). (E) Hyphal length was measured 3 h post infection of oral epithelial cells (n = 3). The difference between Oak 1 and Evo was not statistically significant. (F) Invasion was measured by infecting oral cells, fixing, and differentially staining 3 h post infection. Invasion was calculated as percentage of invading hyphae compared to all counted fungal cells (n = 3). (G) *ECE1* expression was analyzed after 3 h of growth in hypha-inducing medium. The expression is normalized to *ACT1* (ΔCt). The control condition is the yeast morphology of the reference strain SC5314 (ΔΔCt) (n = 3). The difference between Oak 1 and Evo was not statistically significant. (A-G) Significance by one-way ANOVA with Tukey's multiple comparisons test (* p < 0.05, ** p < 0.01, *** p < 0.001). (H) Predicted Ece1 sequences were aligned using Clustal Omega. Identical amino acids are indicated as asterisks (*). Blue indicates the signal sequence, and pink-colored amino acids show the Kex protease cleavage sites of the polypeptide.

We hypothesized that the increased filamentation might have led to enhanced expression of hypha-associated genes [52] in the sugar-evolved strain, contributing to the increased damage. We indeed found increased expression of two aspartic protease genes, *SAP4* and *SAP6*, at both, 24 h and 48 h after infection of intestinal cells, as well as for the adhesin gene *ALS1* at the later time point (S6 Fig). All other hypha-associated genes, however, such as *ALS3*, *ECE1*, or *HWP1* did not differ in their expression. This suggested that the increased damage by the sugar-adapted strain at these later infection stages is mainly driven by its enhanced filamentation and not by an increased transcription of hypha-associated genes. One possible mechanism linking filamentation and invasion to epithelial cell damage is the delivery of the toxin candidalysin into the invasion pocket [49]. We therefore first verified, by qRT-PCR, that there were no differences in *ECE1* transcript levels between the sugar-adapted strain and its parent on oral and intestinal cells at 24 h and 48 h (S6 Fig). We additionally examined the expression of *ECE1* at 3 h, when the *ECE1* transcript levels are generally highest in *C. albicans*, which is known to be necessary for damage to the host [49]. Interestingly, the sugar-adapted strain showed increased *ECE1* transcript levels compared to its parental strain at 3 h in hypha-inducing conditions (Fig 7G). We also investigated the Ece1 peptide sequences and searched for mutations that may have evolved during our sugar evolution experiment, but found no genetic changes (Fig 7H). For the promoter region of *ECE1*, only minor mutations were found (S2C Fig), which did not affect the TATA box at -109 bp, the main cis-factor for *ECE1* transcription in *C. albicans* [47].

In conclusion, adaptation to a dietary sugar led to the emergence of a *C. albicans* strain with increased filamentation on host cells and stronger invasion into the epithelial cell layer.

Higher damage capacity and increased filamentous growth are often associated with an increased ability for translocation through the intestinal epithelium [53,54]. Therefore, we tested the translocation abilities of the different strains using a transwell model with intestinal cells [53]. SC5314 showed the highest translocation 24 h post infection, while Oak 1 showed the lowest and Oak 2 had an intermediate level (Figs 8A and S7A), largely matching our intestinal cell damage findings (Fig 2B). For the sugar-adapted Evo strain, the translocation rate was increased compared to its parental strain, even though the level was still significantly lower than the SC5314 reference strain (Fig 8A). Measurements of the transepithelial electrical resistance (TEER), an indicator for barrier integrity, revealed that SC5314 decreased the overall integrity of the barrier significantly more than either oak tree strain or Evo (Figs 8B and S7B).

To further investigate the increased virulence of Evo in a more complex infection scenario, we used a previously established gut-on-chip model [55]. The *C. albicans* strains were individually used to infect the gut compartment by perfusing them within that compartment in the organ-on-chip. Unadhered CFUs obtained from the flow-through showed that all strains adhered to the gut-side intestinal epithelial cells to a similar degree (Figs 8C and S7C). Damage to the host cells in the gut and vascular compartments was measured by LDH release after 24 h. Surprisingly, the otherwise non-virulent Oak 1 isolate damaged both, intestinal and endothelial cells to the same extent as SC5314 in this setting (Fig 8D). The lowest damage to both the epithelial and endothelial compartment was observed for Oak 2, in stark contrast to its high pathogenicity in the earlier *in vitro* assays (S7D Fig). The sugar-adapted Evo, however, induced damage to the intestinal cells to the same extent as SC5314 and even showed the highest damage of all strains to the vascular cells. This again supports the previously observed increase in its virulence (Fig 8D) and suggests a galactose-driven adaptation to the gut niche, which among our models is best simulated by this organ-on-chip system.

To determine the colonization, invasion, and translocation abilities for each strain in this setting, CFUs recovered from the different compartments of the chip were plated after 24 h. Chips infected with Oak 1 or with Evo showed more CFUs in the gut flow-through. A likely reason for this is that they form more less-adherent yeasts than the more filamentous SC5314 and Oak 2 strain (Figs 8E and S7E). When the total intestinal epithelium was scraped and plated to determine the intestinal fungal burden, the majority of strains showed similar degrees of growth and invasion. Only Evo had a significantly higher fungal load in the intestinal compartment (Fig 8E).

Based on the CFUs levels on the vascular side, all strains had a similar ability to invade the endothelium and translocate. The fungal burdens of the oak tree isolates and Evo strain were even higher than that of SC5314 (Figs 8E and S7E,

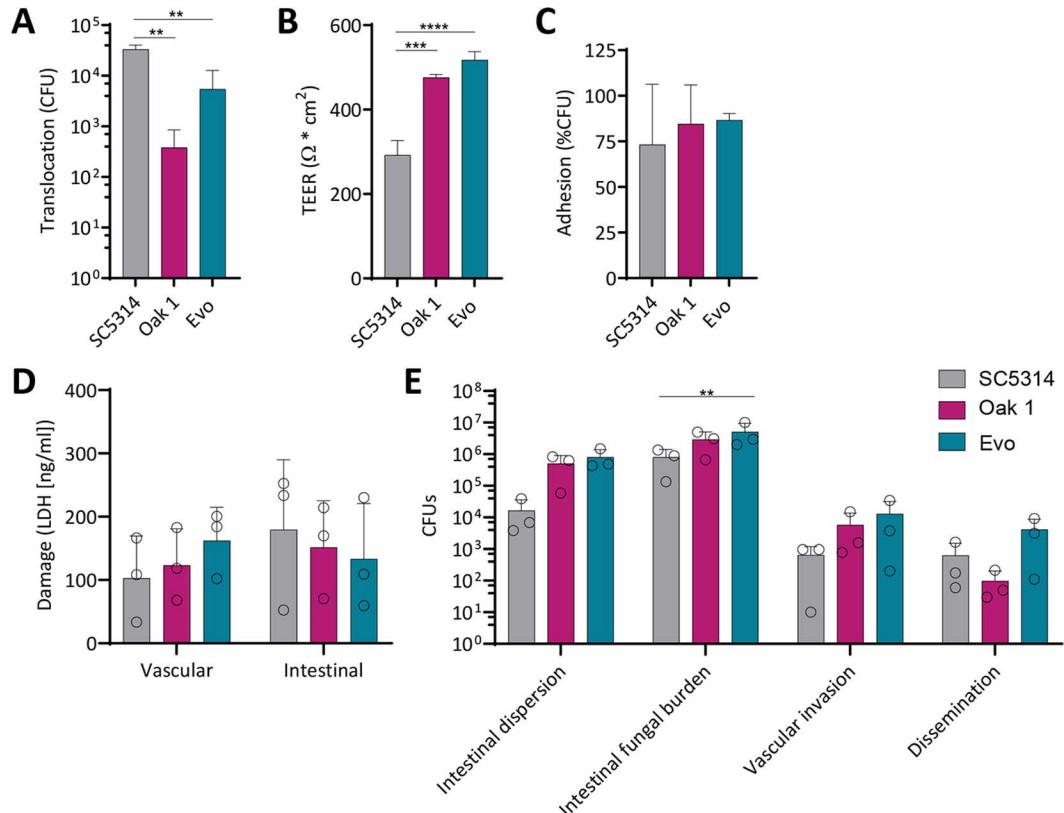

**Fig 8. Virulence of the parental Oak 1 isolate and the sugar-adapted Evo strain in a transwell assay and a complex gut-on-chip model.** (A) Translocation of the strains was measured in an intestinal transwell model by plating the translocated colonies 24 h post infection. Significance was calculated using a one-way ANOVA with Tukey's multiple comparison test (** $p < 0.01$) (n = 3). (B) Barrier integrity was determined by measuring TEER (transepithelial electrical resistance) in an intestinal transwell model 24 h after infection. Significance was calculated using a one-way ANOVA with Tukey's multiple comparison test (*** $p < 0.001$, **** $p < 0.0001$) (n = 3). (C) Percentage of adhesion to the gut epithelium in the gut-on-chip model was determined by plating the amount of non-attached *C. albicans* in the flow-through relative to the injected inoculum (n = 3). (D) Damage was measured as LDH release from both, the gut side and the vascular compartment of the gut-on-chip 24 h post infection (n = 3). (E) Fungal burden was determined by plating the intestinal dispersion (gut flow-through), intestinal fungal burden (intestinal lysate), vascular invasion (vascular lysate), and dissemination (vascular flow-through) 24 h post infection (n = 3). To show the variation between the different gut-on-chip experiments, the individual values of the biological replicates are shown in (D) and (E). Significance was calculated using a two-way ANOVA with Tukey's multiple comparison test (** $p < 0.01$). The difference between Oak 1 and Evo was not statistically significant in panel (A), (B), (C), (D), and (E).

Vascular invasion). Although the translocation into the endothelium was similar for all strains, they differed in the ability to disseminate into the vascular medium (Figs 8E and S7E). The sugar-adapted Evo disseminated better than the Oak 1 strain, matching its increased gut- and vascular-side damage, and its increased translocation in the transwell model (Fig 8A, D, E). Overall, these data show that adaptation of an environmental isolate of *C. albicans* to host dietary sugars can increase its virulence potential in the intestinal niche.

Lastly, we checked whether the sugar adaptation changed *C. albicans* interaction with macrophages. Similar to the parental oak tree isolate, the sugar-adapted Evo strain showed an increased intracellular survival in macrophages compared to SC5314 (S8C, D Fig). Moreover, when investigating the damage induction to human macrophages, the end-point LDH assay showed that Evo still inflicted significantly less damage than SC5314, similar to its parent Oak 1 (S8A Fig). However, when measuring the host cell membrane lysis over time, the sugar-adapted Evo strain induced macrophage lysis to a similar extent than SC5314, significantly more than Oak 1 (S8B Fig).

We investigated the immune activation by the evolved strain in whole blood, with macrophages, or in the complex gut-on-chip model. For the latter, no difference in cytokine release was observed among the strains (S9C Fig). Interestingly, in whole blood, the sugar-adapted Evo induced less pro-inflammatory cytokines such as IL-8, TNFα, IL-6, and IL-1β than SC5314, and less IL-8 than its parental strain (S9A Fig). This trend, however, was inverted in infected macrophages, as Evo induced the highest release of IL-8 and TNFα (S9B Fig).

## Discussion

*Candida albicans* is a common member of the human microbiome and an opportunistic pathogen [56]. There are few studies that have described environmental isolates, and it has been a contentious issue whether environmental *C. albicans* isolates are generally contaminations from human influences [8]. This is especially true as until the 1980s this pathogenic fungus was only rarely isolated from environmental sources, and even then mostly from human-associated sources such as urban aquatic environments, soil, hospitals, and vegetables [9–16]. However, recently multiple *C. albicans* strains have been isolated from non-urban environmental niches [17,18,20,21], which supports the hypothesis of natural habitats of these pathogenic yeasts outside of clinical settings (commented on by Morio *et al.* [22]).

In this study, we investigated clinically relevant phenotypes of three environmental isolates originally found on oak trees [17]. It can be assumed that these isolates have not been in contact with a human host for an extended period of time, if at all, as discussed in their original description by Bensasson *et al.* [17]. We hypothesized that these oak tree isolates would therefore be maladapted to humans and exhibit a lower pathogenic potential compared to clinical isolates. However, our research revealed that one of these oak tree *C. albicans* strains, Oak 2, instead has phenotypes which resemble a highly virulent clinical isolate. Oak 2 showed growth comparable to the host-adapted clinical strain SC5314 and other clinical isolates with the majority of tested carbon and nitrogen sources. In general, all oak tree isolates were not exceptional in their growth pattern, and fell within the phenotypic diversity of the tested clinical isolates. Furthermore, Oak 2 exhibited equivalent or even higher levels of adhesion, invasion, filamentation, and damage to host cells, including epithelial cells and macrophages. This highly virulent phenotype seems to be intrinsic to the fungus, as immune cells reacted similarly to all oak tree isolates, suggesting that it is not due to differences in the host responses. Oak 1 and Oak 3, however, showed lower virulence across all tested host cells, as did most of the clinical isolates we tested. This indicates that the oak tree isolates' virulence potentials are within the phenotypic diversity of human-associated *C. albicans* strains, with Oak 1 and Oak 3 exhibiting characteristics of low-virulence strains and Oak 2 resembling a highly virulent clinical isolate.

Whether some of these virulence phenotypes have evolved due to exposure to predators [6] or are a by-product of other adaptations, *e.g.,* to sugars as in our experimental evolution setup, remains unclear. The concept of an "environmental virulence school" [6] is well-established for other environmental fungal pathogens such as *Cryptococcus neoformans* (reviewed in [57–59]). Several authors have suggested that its interaction with soil amoebae led to the evolution of virulence factors that allow this fungus to survive mammalian macrophages [60,61]. A similar "training" of virulence has been demonstrated for the first time recently in environmental *C. albicans* isolates [21].

However, we did not find differences in survival upon amoeba predation between the highly virulent Oak 2 and the other less virulent isolates, suggesting that the virulence phenotype of Oak 2 cannot simply be explained by adaptation to environmental amoebae. The observed high damage potential may in general rather be attributed to the strong filamentation on host cells, promoting higher adhesion and invasion. Moreover, this could result in an increased expression of hypha-associated genes, including hydrolytic secreted aspartyl proteinases (Saps), specifically Sap4–6, that can contribute to host tissue damage [62]. Among these hypha-associated genes is also *ECE1*, which encodes the cytolytic toxin candidalysin, the key factor responsible for host cell damage by *C. albicans* [42,63]. It is notable that all oak tree isolates code for slightly different candidalysin variants [42], and especially that two of them (in Oak 1 and Oak 2) have not been found before in a larger screen of clinical isolates[46]. Indeed, the 182 strains in this screen shared only 10 variants, which makes the detection of two new sequences in three strains somewhat remarkable. Additionally, we found amino acid

substitutions in the non-candidalysin Ece1 peptides (NCEPs) in the oak tree isolates compared to SC5314. The NCEPs are proposed to prevent auto-aggregation of CaL, which ensures proper Ece1 processing and secretion, thereby affecting the induction of host cell damage [48]. Furthermore, the peptide 2-candidalysin sequence combination is critical for the toxin-associated virulence as amino acid substitutions can lead to lower cleavage and secretion efficiency [45]. Interestingly, the majority of amino acid substitutions was in fact found in peptides 2, 3, and 7 in Ece1 variants of the environmental strains. Moreover, Oak 2 has an altered promoter region, with its TATA box being further upstream. While we cannot be certain about the specific effects of these changes in its regulatory region, this may suggest that this environmental, highly virulent *C. albicans* isolate has evolved a specific, potent *ECE1* variant. Other closely related environmental *Candida* species possess an *ECE1* homologue, and especially synthetic *C. tropicalis* candidalysin is remarkably potent on oral epithelial cells [44,46]. However, *C. tropicalis* does not seem to produce the toxin during *in vitro* host cell infections [44]. In contrast, the Oak 2 isolate expressed *ECE1* on epithelial cells similarly to SC5314 but peaked at a later timepoint, potentially causing greater damage at later stages. This strong *ECE1* expression on oral and intestinal cells, which led to measurable candidalysin secretion, in combination with the high invasion potential on epithelial cells [49], can explain the strong damage induction and virulence potential of Oak 2 on host cells.

Surprisingly, the highly virulent Oak 2 isolate was even resistant to the broad-spectrum antifungal amphotericin B, a property rarely observed even in clinical isolates of *C. albicans* [64], and most environmental fungi are typically susceptible to this antifungal compound [65–67]. However, in the sequences provided by Bensasson *et al.* [17], both *UPC2* alleles of the Oak 2 strain (NCYC 4145) show frameshift mutations that disrupt the C-terminal activation domain of the encoded protein. Upc2 is a key transcriptional regulator of ergosterol biosynthesis, and its deletion is known to markedly reduce ergosterol levels [68,69] – the primary target of amphotericin B. No major mutations are present in several genes of the ergosterol pathway itself, like *ERG11*, *ERG5* or *ERG3*, or in the *MDR1* gene, and the *UPC2* gene is SC5314-like in the other two Oak tree isolates. Consequently, the *UPC2* mutations of Oak 2 may significantly affect its susceptibility to amphotericin B.

Of note, all three oak tree isolates showed a decreased susceptibility to fluconazole. Exposure of environmental fungi to agricultural azoles has been suggested as a cause for resistance development [70,71], and this cannot be excluded here: The isolates were found at isolation sites that are not too far from agriculturally used areas [17]. While this would not explain the amphotericin B resistance, they may also have acquired resistance *via* other selective pressures in the environment such as antifungal metabolites secreted for example by environmental *Streptomyces* species [72–74]. Antifungal resistances often result from biological factors and processes evolved for other purposes, such as efflux pumps against xenobiotics or altered cell wall composition [65,75,76]. Bensasson *et al.* identified a trisomy of chromosome R in the parental oak tree isolate 1 [17], a trisomy that is known to confer resistance to triazoles [77]. Nevertheless, we could not confirm such an aneuploidy in our whole genome sequencing data. Aneuploid chromosomes, however, can easily revert back to the diploid state due to laboratory sub-cultivation [77].

In conclusion, our investigation of three *C. albicans* oak tree isolates identified a highly virulent, antifungal-resistant strain, namely Oak 2, which belongs to a previously undescribed *C. albicans* clade [17]. More importantly, this suggests the possibility of an environmental reservoir of *C. albicans* isolates that are pre-adapted to efficiently colonize and infect the human host. Bensasson *et al.* convincingly argued that the oak tree isolates are of genuinely environmental origin [17]. It is not known, however, whether the strains have been permanently existing far from animals for an evolutionary significant amount of time, or whether they were (transiently and or intermittently) associated with warm-blooded animals such as birds or rodents [17]. This could have affected the pre-adaption of Oak 2 to humans, a process that has been described for other usually non-human-associated pathogenic fungi such as *H. capsulatum* and *C. neoformans* [19,78–80].

While the environmental selective pressure for the virulence and resistance phenotypes of the Oak 2 isolate remains unclear, we used an experimental evolution approach to investigate the effects of human dietary sugars on the virulence of *C. albicans per se*. With this strategy, we aimed to recapitulate, in part, the recent evolutionary adaptation of this fungus to the sugar-rich western diets of its human host [23].

PLOS Pathogens

We used the less virulent *C. albicans* oak tree isolate Oak 1 and serially passaged it in galactose-rich medium. Interestingly, the resulting strain showed drastically increased growth not only with galactose, but also with other dietary sugars, including glucose and fructose [32,33], and even some host-associated amino acids [34]. Moreover, we observed that the evolved strain underwent a transcriptomic remodeling, wherein multiple (sugar) metabolism genes and pathways were differently regulated. This is likely due to the upregulation of multiple transporters responsible for the import of various sugars, as well as in upregulation of glucose and galactose sensors (S4 Fig), although our analysis did not yet pinpoint a clear underlying genetic alteration among the candidate mutations. In consequence, the evolved strain developed more rapid sugar import and catabolism, resulting in an overall increased metabolic flexibility.

As a successful gut colonizer*, C. albicans* requires a high degree of metabolic flexibility and plasticity [34]. This suggests that our galactose-evolved strain should have acquired an enhanced colonizing potential. In fact, the sugar-evolved strain showed a slightly increased fungal burden in the gastrointestinal gut-on-chip model compared to SC5314. An association between diet and *C. albicans* colonization is well-established, as for example a diet rich in sugar increases the number of *C. albicans* cells in the murine gut [81]. Similarly, the consumption of sweetened foods has been shown to promote the colonization of *Candida* spp. in the human gastrointestinal tract [82].

Prior research has also documented the short-term influence of dietary sugars on fungal virulence: Samaranayake *et al*. and McCourtie *et al*. showed that carbohydrates, including glucose, maltose, and galactose affect the adhesion of *C. albicans* on epithelial cells and plastic [83,84]. These short-term metabolic adaptations likely increased *C. albicans*' virulence *via* rapid and transient transcriptional changes. Interestingly, the long-term metabolic rewiring during the sugar adaptation in our study had a similar effect, rendering *C. albicans* more virulent on intestinal epithelial cells. Given the duration of the evolution experiment, which spanned several weeks, it is probable that long-term nutritional remodeling fixated these short-term changes and adaptions, including their effect on fungal virulence. Our whole genome sequencing of the strains did not identify any evident mutation or candidalysin-related change that could explain the enhanced virulence of the sugar-adapted strain. Nevertheless, we showed an increase in hypha formation and invasion on oral epithelial cells, and these phenotypic effects likely contributed to the increased damage to intestinal cells.

In addition to increased metabolic plasticity and virulence potential, serial passaging in this dietary sugar also decreased the strain's susceptibility to amphotericin B and fluconazole. Changes in ergosterol biosynthesis genes such as *ERG11*, *ERG2*, *ERG3*, and *ERG5* have been described as mediators of *C. albicans*' resistance to polyene and azole antifungals [75,85–87]. However, in our RNA-seq dataset we did not observe transcriptional changes for these genes. Nevertheless, we found an upregulation of ABC transporter genes in the sugar-evolved strain compared to its parent, which includes multidrug transporters that are known to confer resistances to azoles [51]. Another mechanism by which catabolic metabolism may be linked to amphotericin B and azole resistance, is through mitochondria, as galactose is known to stimulate mitochondrial functions [88], and the organelle is important for ergosterol biosynthesis [89,90]. Loss of mitochondrial DNA due to ergosterol depletion, or other mitochondrial dysfunctions, including uncoupling of oxidative phosphorylation, can lead to azole resistance in yeast [90–92]. In turn, disrupted mitochondrial function in *C. albicans* results in reduced ergosterol levels and, consequently, decreased susceptibility to amphotericin B [89].

Our study identified two distinct *C. albicans* strains that developed antifungal resistance in the absence of direct antifungal exposure. The first strain acquired resistance during an *in vitro* evolution experiment, which employed host dietary sugars to simulate the gut environment. This finding suggests that evolutionary processes during *C. albicans* gut commensalism may promote antifungal resistance - possibly through exposure to specific dietary carbon sources. In this Evo strain, resistance was associated with alterations in efflux pump expression, an observation that warrants further investigation. In contrast, the environmental Oak 2 isolate demonstrated intrinsic resistance to two classes of antifungals, underscoring that antifungal resistance can emerge through inherent mechanisms.

Collectively, our data suggest an evolution-driven, strong impact of dietary sugars on the virulence potential and antifungal resistance of human pathogenic fungi like *C. albicans*. Hence, sugar-rich diets such as those often consumed in westernized populations could affect colonization, infection, and resistances of *C. albicans* and thereby infection outcomes. However, it is yet unclear whether *C. albicans* must first adapt to dietary compounds in order to gain the ability to colonize and infect the human host or if it adapts to dietary sugars during colonization in the gut. Our findings lend support to both hypotheses, with the highly virulent environmental strain serving as an example for adaptation prior to contact with the human host. The latter hypothesis is supported by the months-long evolution experiment with a sugar present in the gastrointestinal tract, which simulated one possible condition that allows *C. albicans* to evolve within the human host.

Taken together, our study contributes to the evidence suggesting that *C. albicans* may not be exclusively an obligate human commensal [23,24]. Moreover, if an environmental reservoir for *C. albicans* colonization and infection exists undetected – as indicated by the findings of Bensasson *et al*. [17] and others [18–22], along with our own results – it certainly warrants further investigation. In light of this possibility, it becomes intriguing to speculate about the source of each individual's commensal *C. albicans* strains. The prevailing hypothesis posits that humans are initially colonized by *C. albicans* through vertical transmission from a mother to her offspring at the time of birth [93–95]. However, it remains unknown whether humans retain this "original" strain throughout their lifespan, which would result in evolutionary processes to adapt to each individual's dietary habits. In light of our findings, it is plausible that this original *C. albicans* can become more virulent and even antifungal-resistant, depending on the diet. If we consider that each individual acquires a single *C. albicans* strain, this strain would also be the one that, in the event of immunosuppression or microbial imbalance, translocates from the gut and initiates a potential systemic infection. Accordingly, *C. albicans* blood stream infections originate from commensal populations in the gastrointestinal tract [96,97].

Alternatively, or potentially additionally, humans may acquire multiple strains throughout their lifetime, including from the environment. Supporting that, *C. albicans* isolates from remote, closed-off communities do not cluster phylogenetically together, indicating different sources of colonization [24]. Different colonizing strains would likely have different capabilities to cause disease or evade the host immunity and antifungal treatment. Some of these *C. albicans* strains may act as successful commensals, even replacing an original strain, while others will have a pathogenic relationship with their human host. In fact, closely related strains from the same host showed variation in phenotypes, such as filamentation in macrophages and antifungal susceptibility [40]. This suggests that within-host selection, potentially also based on dietary compounds, can result in the emergence of *C. albicans* strains with varying degrees of antifungal resistance and virulence potential from human- or environment-derived progenitors.

## Materials and methods

### Ethics statement

Human peripheral blood was collected from healthy volunteers with ethics approval and after obtaining written informed consent. This study was conducted according to the principles of the Declaration of Helsinki. The blood donation protocol and use of blood for this study were approved by the institutional ethics committee of the University Hospital Jena (permission number 2207–01/08).

### Fungal strains and culture conditions

*C. albicans* SC5314 [98], *C. albicans* oak tree isolates 1, 2, and 3 [17,20], and a sugar-adapted *C. albicans* strain were used for this publication (S1 Table: Strains used in this study). For all experiments single colonies were picked from Yeast Extract Peptone Dextrose (YPD) agar plates and grown overnight in liquid YPD medium in an orbital shaker at 180 rpm at 30 °C. Yeast cells were then harvested by centrifugation (20,000 *g*, 1 min), washed twice with phosphate-buffered saline (PBS), and the yeast cell number was adjusted.

## Culture of oral and intestinal cells

The human oral epithelial cells (TR146, European Collection of Authenticated Cell Cultures ECACC #10032305) were cultivated at 37 °C and 5% $CO_2$ in Dulbecco's modified Eagle medium/Nutrient Mixture F-12 (DMEM/F12, Life Technologies) supplemented with 10% fetal bovine serum (FBS) (Bio&Sell) for no longer than 15 passages. TR146 cells were seeded in 6-well plates at a concentration of $8 \times 10^5$ cells/well for RNA isolation. For damage assays TR146 cells were seeded at a total concentration of $2 \times 10^4$ cells/well in a 96-well plate. For invasion/adhesion/ filamentation assays $1 \times 10^5$ TR146 cells/well were seeded on a coverslip in a 24-well plate. Confluent TR146 cells were washed once and subsequent experiments were carried out in serum-free medium.

The intestinal epithelial Caco-2 brush border expressing 1 cell line (C2BBe1; ATCC, CRL2102) and the human intestinal goblet cell line (HT29-MTX; ATCC, HTB-38; CLS, Lot No. 13B021) were routinely cultivated in Dulbecco's Modified Eagle's Medium (DMEM) (Gibco, Thermo Fisher Scientific) supplemented with 10% fetal bovine serum (FBS) (Bio&Sell), 10 µg/ml Holotransferrin (Calbiochem, Merck), and 1% non-essential amino acids (Gibco, Thermo Fisher Scientific) at 37 °C with 5% $CO_2$ for no longer than 15 passages. C2BBe1 cells were seeded in 6-well plates at a concentration of $5 \times 10^5$ cells/well for RNA isolation. C2BBe1 and HT29-MTX cells were seeded in 96-well plates and transwell inserts (polycarbonate membrane with 5 µm pores; Corning) at a 70:30 ratio (C2BBe1:HT29-MTX) and a total concentration of $2 \times 10^4$ cells/well or insert for damage and translocation assays, respectively. For quantification of filamentation and invasion, intestinal cells were seeded in a 70:30 ratio (C2BBe1:HT29-MTX) and a total concentration of $1 \times 10^5$ cells/well on coverslips. All well plates and transwell inserts were coated with collagen I (10 µg/ml for 2 h at room temperature; Thermo Fisher Scientific) and maintained for 12 d at 100% confluency for differentiation with regular medium exchange before infection. Just prior to infection with *C. albicans*, the medium was removed and fresh DMEM without FBS, Holotransferrin, or 1% non-essential amino acids was added to the cells.

## Growth curves

Washed *Candida* cells were diluted to an $OD_{600\,nm}$ of 0.1. As medium, 0.67% yeast nitrogen base (YNB, BD Bioscience) was supplemented with 2% of a carbon and 0.5% of a specific nitrogen source (for amino acids 10 mM was used). For all tested carbon sources, 0.5% ammonium sulfate was used as nitrogen source and for all tested nitrogen sources, 2% glucose as carbon source. Growth was monitored in 96-well-plates by measuring the absorbance at 600 nm every 15 min for up to 72 h at 30 °C in a microplate reader (Tecan M-Plex). Prior to each measurement plates underwent ten seconds orbital shaking followed by ten seconds waiting time. The first measured $OD_{600\,nm}$ was subtracted, the remaining measured values were plotted, and the area under the curve was calculated. The experiment was repeated three times (n = 3).

## Antifungal drop tests and growth curves

For the drop test, 5 µl of serial diluted *C. albicans* solution were dropped on YPD plates supplemented with the indicated concentrations of amphotericin B (Sigma) and fluconazole (Sigma), and were incubated at 37 °C for 6 days. For the growth curves, YPD was supplemented with either 0.5 µg/ml and 1 µg/ml of amphotericin B, or 32 µg/ml and 64 µg/ml of fluconazole and growth was measured and analyzed as described above.

## Amoebae infections

*Protostelium aurantium* var. *fungivorum* [99] was cultured as described previously [100]. The amoeba trophozoites were regularly grown in standard-size petri dishes (Greiner Bio-One) in 2 mM phosphate buffer (PB) (0.8 mM $KH_2PO_4$, 1.2 mM $K_2HPO_4$, pH 6.6) with *Rhodotorula mucilaginosa* as a food source at 22 °C. The survival of the *C. albicans* oak tree strains in presence of the amoebae was determined as described previously [100]. Briefly, amoebae were grown to confluency in 20 × wMY (40 mg/ml yeast extract, 40 mg/ml malt extract, 0.75 g/l $K_2HPO_4$), harvested by

scraping, and counted. *C. albicans* cultures were prepared as indicated above, washed twice in 20 × wMY, and added to the amoebae at an MOI of 10 in a 96-well plate. After an incubation step at 22 °C for 3 h, the samples were appropriately diluted in PB and plated on YPD plates. The number of surviving yeasts was calculated as a percentage of CFUs relative to the inoculum. To assess the amount of phagocytosed fungi, *C. albicans* and amoebae were prepared as described above and were added in a ratio of 10:1 (fungus:amoeba) in a 24-well plate. After an incubation step for 2 h at 22 °C, samples were fixed with Roti-Histofix 4%. After rinsing with PBS, amoebae were permeabilized with 0.5% Triton X-100 for 10 min. Fungal cells were stained with calcofluor white (CFW, Sigma) and after subsequent washing steps with ddH$_2$O, amoeba actin was stained with Phalloidin-AlexaFluor 633 (Thermo Fisher Scientific). Micrographswere taken using a Zeiss Celldiscoverer 7 (Zeiss, with LSM 900) and the number of amoebae with ingested yeasts (%) was determined.

### Quantification of adhesion, invasion, hypha formation, and hyphal length

Oral cells were infected in a 24-well plate with *Candida* cells at a multiplicity of infection (MOI) of 1 as described above, and incubated for 1 (adhesion) or 3 h (invasion, hypha formation, hyphal length). Non-adherent *Candida* cells were removed by rinsing with PBS and samples were fixed with Roti-Histofix 4% (Roth). To measure adhesion, adherent *Candida* cells were stained with AlexaFluor647 conjugate of succinylated concanavalin A (ConA; Invitrogen) and visualized using a fluorescence microscope (Leica DM5500B, Leica DFC360 FX). Six representative images were counted, the average amount of cells per image was calculated, and the numbers were plotted relative to the reference strain (SC5314). For invasion, extracellular, non-invasive fungal components were stained with ConA. After rinsing with PBS, oral cells were permeabilized with 0.5% Triton X-100 for 10 min. The entire fungal cells (invasive and noninvasive) were stained with Calcofluor white (CFW; Sigma-Aldrich) and visualized by fluorescence microscopy. The total hyphal length was measured, and the percentage of filaments and invasive hyphae (only CFW-stained), counted from at least 100 hyphae per strain for each biological replicate.

Intestinal cells were infected at a MOI of 1 and incubated for 6 h for quantification of filamentation and invasion. Non-adherent *Candida* cells were removed by rinsing with PBS and samples were fixed. Extracellular hyphae were stained with CFW, and the host cells were permeabilized as stated above. Intracellular *C. albicans* hyphae were incubated with 25 µg/ml rabbit anti-*C. albicans* antibody (Acris) for 3 h at 4 °C. After rinsing with PBS, 4 µg/ml goat anti-rabbit antibody labeled with AlexaFluor488 were added, and incubated for 1 h at 37 °C.

### Epithelial damage assay with oral and intestinal cells

Oral and differentiated intestinal cells were seeded in a 96-well plate as described above, infected with *C. albicans* (MOI 1), and incubated for 24 or 48 h. Release of the cytoplasmic enzyme lactate dehydrogenase (LDH) was measured as a marker for necrotic epithelial damage [101] using a Cytotoxicity Detection Kit (Roche) according to the manufacturer's instructions. The LDH release was expressed as % of full lysis control (maximum LDH release induced by the addition of 0.5% Triton X-100 to uninfected epithelial cells) unless otherwise stated.

### Translocation assay

Intestinal cells were seeded in inserts in a 24-well plate as described above. The infection and read-outs were performed as described previously [53]. Briefly, the inserts were infected with *C. albicans* (MOI 1) and incubated at 37 °C and 5% $CO_2$. After 24 h, the translocated hyphae were detached by treating in the lower compartment with 20 U/ml zymolyase (Amsbio) for 2 h at 37 °C and 5% $CO_2$. Detached hyphae were then plated on YPD agar, and the number of colony-forming units (CFUs) was determined.

The trans-epithelial electrical resistance (TEER) was determined using a volt-ohm meter (EVOM2, World Precision Instruments) after 24 h post infection.

## RNA isolation

For fungal RNA extraction on epithelial cells, oral and intestinal cells were seeded in 6-wells as described above. On the day of infection, media in each well was replaced with 2 ml DMEM (intestinal) or DMEM F12 (oral cells) without FBS, infected with *C. albicans* cells (2 ml of $4 \times 10^5$ yeast/ml in medium without FCS), and incubated at 37 °C, 5% $CO_2$. Samples for RNA isolation were collected at different time points: 24 and 48 h post infection. More specifically, the well content was removed and replaced with 500 µl of RNeasy Lysis (RLT) buffer (Qiagen), containing 1% β-mercaptoethanol (Roth). Cells were detached using a cell scraper (< 3 min), immediately shock-frozen in liquid nitrogen, and stored at -80 °C until further use. Collected samples were defrosted and centrifuged for 10 min (20,000 *g*, 4 °C). Fungal RNA was isolated from the pellet, using a freezing-thawing method, as described previously [102]. RNA concentrations were quantified using a NanoDrop 1000 Spectrophotometer (Thermo Fisher Scientific) and RNA quality was assessed with an Agilent 2100 Bioanalyzer (Agilent Technologies).

To compare *ECE1* expression levels during filamentation of the oak tree isolates, *C. albicans* strains were adjusted to $10^7$ cells/ml in 25 ml RPMI 1640 (hypha-inducing) or 5 ml YPD (yeast-maintaining). For hyphae samples, fungal suspensions were distributed in 150 cm² petri dishes and incubated at 37 °C and 5% $CO_2$ for 3 h. Afterwards, medium and nonadherent *Candida* cells were discarded. Adherent *Candida* cells were washed once with ice-cold PBS, detached using a cell scraper, and collected. For yeast samples, fungal suspensions were incubated in Erlenmeyer flasks at 30 °C and 180 rpm for 3 h. The cells were collected at 4000 *g* for 2 min at 4 °C and resuspended in 10 ml ice-cold PBS. Both, yeast and hyphae samples, were centrifuged again at 3000 *g* for 2 min at 4 °C, the supernatant was discarded, and the *C. albicans* pellets were shock-frozen in liquid nitrogen. Thawed pellets were resuspended in 600 µl RNeasy Lysis (RLT) buffer (Qiagen) and RNA was isolated as described previously [49].

## Reverse transcription-quantitative PCR (RT-qPCR)

Isolated RNA (500 ng) was treated with DNase I (Fermentas) following the manufacturer's recommendations and subsequently transcribed into cDNA using 0.5 µg Oligo(dT)12–18 Primer, 200 U Superscript III Reverse Transcriptase and 40 U RNaseOUT Recombinant RNase Inhibitor (Thermo Fisher Scientific). Obtained cDNA was diluted 1:5 and used for qPCR with GoTaq qPCR Master Mix (Promega) in a CFX96 thermocycler (Bio-Rad). The expression levels were normalized against beta-actin. All primers used are listed in S2 Table.

## RNA sequencing and transcriptional analysis

RNA sampling and isolation was performed from intestinal cell infections as described above. Library preparation and RNA sequencing were carried out at Genewiz/Azenta (Leipzig, Germany). After poly(A) filtering, mRNA was fragmented, and cDNA libraries were generated for each sample. Libraries were sequenced with 2x150 read lengths using an Illumina NovaSeq platform. Reads were aligned to the SC5314 standard genome (version A22-s07-m01-r146 [103]) using bowtie2 v2.5.2 [104] with standard settings. Transcripts were counted by Subread featureCounts v2.0.5 [105] followed by median ratio normalization.

Next, we performed gene set enrichment analysis and visualized clustering patterns of our data across different strains and conditions using R (v4.3.2). To visualize the effect of different strains and time on gene expression, first principal component analysis was performed using the prcomp function in the stats package (v4.3.2). The first two components were selected and k-means clustering with two centers was performed using factoextra package (v1.0.7). For KEGG pathway enrichment analysis, up- and down-regulated genes were defined with a $\log_2$ fold-change threshold of ±1.5. Significantly enriched pathways (p < 0.05) were identified using the compareCluster function in the ClusterProfiler package (v4.10.1) [106]. A subset of pathways of interest was manually curated and visualized.

## Ece1 detection in hyphal supernatants by LC-MS/MS analysis

Analysis of hypha-secreted Ece1 peptides was optimized for the detection of candidalysin and has been published recently [48]. Briefly, *C. albicans* strains were incubated for 18 h under hypha-inducing conditions. Secreted peptides

were enriched from culture supernatants by solid-phase extraction, dried in a vacuum concentrator, dissolved in 0.2% formic acid in 71:27:2 (vol/vol/vol) acetonitrile (ACN)-$H_2O$-dimethyl sulfoxide (DMSO), and passed through a 10-kDa molecular mass cut-off filter in order to remove intact proteins. LC-MS/MS analysis was carried out on an Ultimate 3,000 nano RSLC system (Thermo Fisher Scientific) coupled to a QExactive HF mass spectrometer (Thermo Fisher Scientific). Peptide separation was performed on an Accucore C4 column (75 µm I.D. × 150 mm, 2.6 µm). Proteome Discoverer 3.0 (Thermo Fisher Scientific) and the Sequest HT algorithm were used to search against the protein database of *C. albicans* SC5314 (https://www.uniprot.org/proteomes/UP000000559; 2023/05/03). Mass spectra were searched for both unspecific cleavages (no enzyme) and tryptic peptides up to four missed cleavages. Precursor and fragment mass tolerances were 10 ppm and 0.02 Da, respectively. A strict false discovery rate (FDR) < 1% (Target Decoy PSM validator node) and a Sequest score (cross correlation, Xcorr) >3 were required for positive protein/peptide hits. Only rank 1 proteins and peptides of the top scored proteins were counted. Label-free protein quantification was based on the Minora algorithm of PD3.0 using the precursor abundance based on intensity and a signal-to-noise ratio >5.

## Monocyte isolation from buffy coats and macrophage differentiation

Preparation of human monocyte-derived macrophages (hMDMs) was performed as described before [107] based on selection of monocytes by magnetic automated cell sorting of CD14 positive monocytes and a differentiation period of seven days. Adherent hMDMs were detached with 50 mM EDTA in PBS and seeded in 96-well plates ($4 \times 10^4$ hMDMs/well) in RPMI 1640 (Gibco, Thermo Fisher Scientific) supplemented with 10% FBS and 50 ng/mL M-CSF (ImmunoTools) and incubated overnight. Before infection with *C. albicans*, the hMDMs were washed once with PBS, and RPMI without FBS was added.

## Whole blood fungal survival

Human whole blood was freshly drawn from healthy volunteers in anticoagulation tubes with recombinant Hirudin (Sarstedt). *C. albicans* overnight cultures were prepared as described above, $10^6$ yeasts were added to 490 µl whole blood, and incubated at 37 °C under gentle rolling of the tubes. Samples were taken after 15 min, 30 min, 60 min, 120 min, and 240 min and plated on YPD in appropriate dilutions in PBS. The plates were incubated at 30 °C and colony-forming units (CFUs) were determined after 2 d to assess fungal survival.

## Macrophage infections

*C. albicans* strains were prepared as described above and hMDMs were infected with a MOI of 5. To assess damage, LDH release from the macrophages was measured with a Cytotoxicity Detection Kit (Roche) after 24 h of infection. To determine intracellular fungal survival 3 h post infection, the supernatant was removed and macrophages were lysed by adding 200 µl $ddH_2O$, scraping the well, and pipetting up and down to break the cells. This was repeated with each well five times. The supernatant and lysate were each diluted with PBS and plated on YPD plates to determine the CFUs. To evaluate the morphology of the *C. albicans* oak tree isolates inside the macrophages, micrographs were taken using a Zeiss Celldiscoverer 7 (Zeiss, with LSM 900) after 3 h of infection.

## Micro-evolution experiment

*C. albicans* oak tree isolate 1 (NCYC 4144) and the reference strain SC5314 were prepared as described above. The washed overnight cultures were diluted 1:500 in the YNB + 0.5% ammonium sulfate + 2% galactose, and incubated at 180 rpm and 30 °C. The strains were passaged every two to three days into fresh medium for 6 weeks. Growth of the strains was measured as $OD_{600\,nm}$ using Aquila biolabs sensor plates (Aquila biolabs GmbH). After each passage, 750 µl of the cultures were mixed with 1 volume of 60% glycerol, shock-frozen in liquid nitrogen, and stored at -80 °C for further analysis. The sugar-adapted Evo strain used in this study was collected with the last passage.

## Genomic DNA isolation and whole genome sequencing

For DNA extraction, *C. albicans* cultures were grown in YPD for 16 h at 30 °C and 180 rpm. The cultures were centrifuged for 5 min at 4000 rpm, resuspended in 1 ml ddH$_2$O, and centrifuged for 2 min at 13000 rpm. The pellet was resuspended in 1 ml lysis buffer (1 M sorbitol, 100 mM sodium citrate, 50 mM EDTA, 0.6 mg/ml lyticase, 2.5% β-mercaptoethanol), incubated at 37 °C for 45 min, and centrifuged for 5 min at 13000 rpm. The pellet was resuspended in 800 µl of protein-ase buffer (10 mM Tris-Cl, 50 mM EDTA, 0.5% SDS, 1 mg/ml proteinase K) and incubated at 60 °C for 30 min. 800 µl of phenol:chloroform:isoamylalcohol (25:24:1) were added, the samples were vortexed for 4 min, and centrifuged for 3 min at 13000 rpm. The aqueous phase was transferred to a fresh tube, 700 µl ice-cold isopropanol were added, and the precipitated DNA was pelleted by centrifugation for 5 min at 13000 rpm. The pellets were washed with 70% ethanol and centrifuged for 2 min at 13000 rpm. After air-drying, the pellet was resuspended in water supplemented with RNase. The gDNA was stored at -20 °C until sequencing. The sequencing was carried out by Genewiz/Azenta (Leipzig, Germany) with 2 × 150 read lengths and 10 M raw paired-end reads using an Illumina NovaSeq platform. Whole genome sequences were used for variant detection followng the best practices workflows as outlined by the GATK Team (gatk.broadinstitute.org, [108]). Briefly, the reads were pre-processed using Picard v3.1.1 [109] to remove any adapter sequences. Reads were then aligned to the NCYC4144 haploid genome [110] using the bwa v0.7.17 BWA-MEM algorithm (with parameters -M -t 11 -B 3 -p). Duplicate reads were removed using Picard in MarkDuplicates mode and the resulting SAM files sorted and indexed. Variants were detected by comparison to the re-sequenced NCYC4144 strain using GATK v4.5.0.0 with mutect2, followed by FilterMutectCalls with standard settings and SelectVariants to exclude the filtered variants.

## Ece1 amino acid sequence analysis

To compare the sequences of the Extent of cell elongation 1 polyprotein in the oak tree *C. albicans* strains and in the sugar-adapted Evo strain, DNA sequences were extracted from whole genome data or for SC5314 from candidagenome.org and the amino acid sequence was predicted using Expasy (Swiss Institute of Bioinformatics). The amino acid sequences were aligned with Clustal Omega (European Bioinformatics Institute). Asterisks indicate identical amino acids and dots structurally similar amino acids. Similarly, promoter regions 400 bp upstream of the ATG from *ECE1* were retrieved either from candidagenome.org or the WGS and aligned with Clustal Omega.

## Gut-on-chip infection model

The gut-on-chip model was performed as described previously [55,111]. Briefly, biochips were prepared, assembled [111], and infected with 1250 fungal cells per chip under linear perfusion. After 10 min, the intestinal compartment was switched to linear perfusion with fresh medium. The unadhered fungal cells were collected for a total of 50 min and plated as a measurement of adhesion to the gut compartment. After 24 h post infection, damage induction, membrane permeability, and fungal burden were measured as described before [111]. For cytokine levels, supernatants were collected from vascular and intestinal compartments 24 h post infection, centrifuged, and stored at -20 °C.

## Cytokine release quantification

Supernatants of infected primary macrophages (MOI 5) were collected 24 h post infection. Whole blood supernatants were collected 4 h post infection after centrifuging samples for 10 min at 500 *g*. Gut-on-chip supernatants were collected as described above. The release of IL-8, IL-6, Il-1β, and TNFα was measured using commercially available human enzyme-linked immunosorbent assay kits (R&D Systems) according to the manufacturer's protocols.

## Statistics and reproducibility

Experiments were performed in at least biological triplicates (n = 3). For experiments using primary macrophages or whole blood, at least 4 different donors were used. Data were analyzed using GraphPad Prism 9.5.1 (GraphPad Software, La

Jolla California USA). Values are presented as mean ± standard deviation (SD). Statistical tests are indicated in each figure legend. When applicable, statistical comparison was always performed between the parent strain Oak 1 and the sugar-evolved Evo strain. Statistical significance is indicated in the figures as follows: *, $p \leq 0.05$; **, $p \leq 0.01$; ***, $p \leq 0.001$; ****, $p \leq 0.0001$.

## Supporting information

**S1 Table. Strains used in this study.**
(DOCX)

**S2 Table. Primers used for RT-qPCR.**
(DOCX)

**S3 Table. Mutations found during comparison of the sugar-adapted Evo strain with its parental oak tree isolate 1.**
(XLSX)

**S1 Fig. Growth of all included clinical isolates and survival of the oak tree isolates upon amoeba predation.** Growth was measured as growth curves at 30 °C in different media comprising specific carbon (A) or nitrogen (B) sources. The growth is shown as area under the curve relative to the reference strain SC5314 (% growth). Asterisks indicate significance compared to SC5314 calculated using a one-way ANOVA with Dunnett's multiple comparisons test (* $p < 0.05$, ** $p < 0.01$, *** $p < 0.001$, **** $p < 0.0001$) (n=3). (C) Survival of *C. albicans* yeast cells after 3 h of co-incubation with *P. aurantium*. Results were compared using a one-way ANOVA with Tukey's multiple comparisons test (n=3). (D) Percentage of *P. aurantium* cells containing internalized *C. albicans* yeast cells after 2 h. Results were compared using a one-way ANOVA with Tukey's multiple comparisons test (* $p < 0.05$) (n=3).
(TIF)

**S2 Fig. *ECE1* promoter region and candidalysin variants.** (A-D) The promoter region is shown 400 bp upstream of *ECE1*. Pink highlights the first TATA box and blue the second, less important TATA box as observed in [47]. Orange indicates a newly identified TATA box in the sequence. Alignments were made using Clustal Omega. Identical amino acids are indicated as asterisks (*). (E) Candidalysin sequences from oak tree isolates compared to clinical isolate sequences (taken from [46]). Different amino acids are indicated in purple. Oak 1 and Oak 2 have unique candidalysin sequences, Oak 3 has the same sequence as variant F, previously found in the clinical strains.
(TIF)

**S3 Fig. Sugar-evolved strain characterization.** The strains Evo 1, Evo 2, and Evo 3 are the triplicates kept during the sugar micro-evolution experiment and are all derived from Oak 1. The strain further characterized in this study is Evo 3, and was named 'Evo' to simplify the nomenclature. (A, B) Antifungal resistance of the galactose-evolved strains was assessed by performing growth curves with YPD supplemented with either 32 µg/ml fluconazole or 0.5 µg/ml amphotericin B at 30 °C for 72 h, and is depicted as area under the curve (n=3). (C) Damage to oral cells and (D) intestinal cells was evaluated by measuring the LDH release 48 h post infection (oral n=3, intestinal n=4). The lower dashed line indicates the uninfected control. (E) Hypha formation was determined by microscopic evaluation counting the number of hypha-forming cells relative to all present *C. albicans* cells on oral cells 3 h post infection (n=3). (F) Adhesion of the *C. albicans* strains was assessed by fixing and staining the fungal cells 1 h post infecting oral cells. The average number of attached *C. albicans* fungal cells per microscopic image is plotted relative to the reference strain SC5314 (n=3). (A-F) For all graphs, significances were calculated by using a one-way ANOVA with Tukey's multiple comparisons test (* $p < 0.05$, ** $p < 0.01$, *** $p < 0.001$). Data of all graphs are depicted partially in the main Figs 5 and 7. For panel (C), (D), and (F), the difference between Oak 1 and Evo was not statistically significant.
(TIF)

**S4 Fig. Gene expression of genes in the sugar metabolism pathways.** (A, B) Galactose metabolism (A) as well as starch and sucrose metabolism KEGG pathways (B) of the sugar-evolved strain relative to its parental oak tree isolate 1. Gene expression was assessed 24 h p.i. on intestinal epithelial cells. Fold change is shown with the indicated color scale. Genes which are depicted in grey were not included in the analysis. (C) $Log_2$(fold change) of genes involved in glucose and galactose sensing, transport, as well as in the downstream signaling pathways [112] of the sugar-adapted strain relative to its parental strain (oak tree isolate 1) on intestinal epithelial cells. Gene functions were taken from candidagenome. org.
(TIF)

**S5 Fig. Filamentation, invasion, and hyphal length of the *C. albicans* oak tree isolates and the sugar-adapted strain on intestinal epithelial cells.** (A) Hypha formation was determined by microscopic evaluation counting the number of hypha-forming cells relative to all present *C. albicans* cells 6 h post infection (n = 3). (B) Hyphal length was measured after 6 h after infecting intestinal cells (n = 3). (C) Invasion was measured by infecting intestinal cells, fixing, and differentially staining 6 h post infection. Invasion was calculated as percentage of invading hyphae compared to all counted fungal cells. Statistical significance was calculated using a one-way ANOVA with Tukey's multiple comparisons test (**** $p < 0.0001$) (n = 3). The difference between Oak 1 and Evo was not statistically significant.
(TIF)

**S6 Fig. Hypha-associated gene expression of the oak tree isolates and the sugar-adapted strain on epithelial cells.** (A) $Log_2$(fold change) of some hypha-associated genes for the non-virulent oak tree isolate 1, the highly virulent oak tree isolate 2, and the sugar-adapted strain (Evo) relative to SC5314 24 h or 48 h post infection of intestinal epithelial cells. (B) $Log_2$(fold change) of some hypha-associated genes of the sugar-adapted strain relative to its parental strain (oak tree isolate 1). For both (A, B), expression was assessed *via* RNA sequencing. (C, D) *ECE1* expression of the sugar-adapted strain on intestinal (C) and oral epithelial (D) cells. Expression was assessed *via* qPCR and is shown as fold change relative to SC5314 at the indicated time points. The expression is normalized to *ACT1*. Statistical significance was calculated using a one-sample *t*-test comparing the mean of the samples with a hypothetical mean of 1 (** $p < 0.01$, **** $p < 0.0001$) (n = 3). The difference of Oak 1 and Evo was not statistically significant.
(TIF)

**S7 Fig. Virulence of Oak 2 compared to SC5314 in a transwell assay and a complex gut-on-chip model.** (A) Translocation of the strains was measured in an intestinal transwell model by plating the translocated colonies 24 h post infection. Significance was calculated using a Welch's *t*-test (* $p < 0.05$) (n = 3). (B) Barrier integrity was determined by measuring TEER (transepithelial electrical resistance) in an intestinal transwell model 24 h after infection. Significance was calculated using a Welch's *t*-test (* $p < 0.05$) (n = 3). (C) Percentage of adhesion to the gut epithelium in the gut-on-chip model was determined by plating the amount of non-attached *C. albicans* in the flow-through relative to the injected inoculum (n = 3). (D) Damage was measured as LDH release from both, the gut side and the vascular compartment of the gut-on-chip 24 h post infection (n = 3). (E) Fungal burden was determined by plating the intestinal dispersion (gut flow-through), intestinal fungal burden (intestinal lysate), vascular invasion (vascular lysate), and dissemination (vascular flow-through) 24 h post infection (n = 3). To show the variation between the different gut-on-chip experiments, the individual values of the biological replicates are shown in (D) and (E).
(TIF)

**S8 Fig. Interaction of sugar-adapted strain with macrophages.** (A) Damage to monocyte-derived macrophages was determined after 24 h by measuring LDH release. Each dot represents one donor. Significance was calculated by using a one-way ANOVA with Tukey's multiple comparisons test (** $p < 0.01$, *** $p < 0.001$) (8 donors). The difference of Oak 1 and Evo was not statistically significant. The lower dashed line indicates the uninfected control. (B)

Host cell lysis of monocyte-derived macrophages was measured by propidium iodide staining over the course of 24 h. Significance was calculated using a two-way ANOVA with repeated measures and Tukey's multiple comparison test (* p < 0.05) (4 donors). (C) Intracellular survival in human monocyte-derived macrophages of the *C. albicans* strains was assessed 3 h post infection by lysing and plating the intracellular fungal cells. Survival is depicted relative to the reference strain SC5314. Each dot represents one donor. Significance was calculated by using a one-sample *t*-test comparing the mean of the samples with 100% (* p < 0.05) (9 donors). The difference of Oak 1 and Evo was not statistically significant. (D) Fungal cells per macrophage were determined 3 h post infection by staining fungal cells (ConA-AlexaFluor647, blue) and cell nuclei (DAPI, purple). Images were taken at 63 × magnification with immersion oil (scale bar, 20 μm), and fungal cells per macrophage were counted. Each dot represents one macrophage. Infected macrophages from 4-6 donors were counted. Significance was calculated by using a one-way ANOVA with Tukey's multiple comparisons test (* p < 0.05, **** p < 0.0001). (E) Phase contrast microscopic pictures were taken 3 h post infecting human monocyte-derived macrophage at 10x magnification, and microscopy images were taken from quantification in (C) at 63 × magnification with immersion oil (scale bar, 20 μm). Samples were stained with DAPI (nuclei staining, purple) and Concanavalin A-AlexaFluor647 (fungal staining, blue). The pictures for SC5314, Oak 1, Oak 2, and Oak 3 are also shown in Fig 4.
(TIF)

**S9 Fig. Immune activation by the *C. albicans* oak tree isolates and sugar-adapted strain.** Release of IL-8, TNF-α, IL-1β, and IL-6 was determined in supernatants of (A) infected whole blood after 4 h of infection (6 donors), (B) by infected primary human macrophages 24 h post infection (5 donors), and (C) in supernatants of the vascular and gut compartment of the organ-on-chip model 24 h post infection (n = 3). Significance in (B) was calculated by using a one-way ANOVA with Dunnett's multiple comparisons test (* p < 0.05). The difference of Oak 1 and Evo was not statistically significant.
(TIF)

## Acknowledgments

We would like to thank Douda Bensasson, University of Georgia, for providing the oak tree isolates. We thank the anonymous blood donors, as well as Sophie Austermeier for organizing and supporting us with the whole blood model and macrophage work. In addition, we would like to thank our lab technicians for supporting with buffy coat isolations and for generating the Ece1 secretion samples for LC-MS/MS. Furthermore, we would like to thank Luisa Fischer, Anna Katharina Renner, and Priti Sai Mundrati for helping with the endless labeling of plates and tubes, as well as helping with microscopic evaluation and plating experiments. We would like to thank Selene Mogavero for providing help and insight with the *ECE1* part of the project and Marisa Valentine for her advice for using the organ-on-chip model. Finally, we want to thank Mark S. Gresnigt and Ger van Zandbergen for fruitful discussions, as well as Lydia Kasper and Verena Trümper for help with experimental questions and for providing their unconditional support.

## Author contributions

**Conceptualization:** Theresa Lange, Sascha Brunke, Bernhard Hube.

**Formal analysis:** Theresa Lange, Raghav Vij, Sascha Brunke, Jakob L. Sprague.

**Funding acquisition:** Falk Hillmann, Axel A. Brakhage, Sascha Brunke, Bernhard Hube.

**Investigation:** Theresa Lange, Silvia Radosa, Jakob L. Sprague.

**Methodology:** Theresa Lange, Raquel Alonso-Roman, Thomas Krüger, Olaf Kniemeyer.

**Resources:** Falk Hillmann.

**Supervision:** Jakob L. Sprague, Falk Hillmann, Stefanie Allert, Sascha Brunke, Bernhard Hube.

**Visualization:** Theresa Lange.

**Writing – original draft:** Theresa Lange.

**Writing – review & editing:** Theresa Lange, Jakob L. Sprague, Stefanie Allert, Sascha Brunke, Bernhard Hube.

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
