## [Decision Letter · Decision Letter 0]

1 Jun 2025

“Pour Some Sugar on Me” – Environmental *Candida albicans* Isolates and the Evolution of Increased Pathogenicity and Antifungal Resistance through Sugar Adaptation

PLOS Pathogens

Dear Dr. Hube,

Thank you for submitting your manuscript to PLOS Pathogens. After careful consideration, we feel that it has merit but does not fully meet PLOS Pathogens's publication criteria as it currently stands. Therefore, we invite you to submit a revised version of the manuscript that addresses the points raised during the review process.

Please submit your revised manuscript within 60 days Jul 31 2025 11:59PM. If you will need more time than this to complete your revisions, please reply to this message or contact the journal office at plospathogens@plos.org. Please include the following items when submitting your revised manuscript:

We look forward to receiving your revised manuscript.

Kind regards,

James B. Konopka

Academic Editor

PLOS Pathogens

Debra Bessen

Section Editor

Editor-in-Chief

PLOS Pathogens

orcid.org/0000-0003-2946-9497

Editor-in-Chief

PLOS Pathogens

orcid.org/0000-0002-7699-2064

**Additional Editor Comments :**

The reviewers thought the topic of your manuscript was interesting, but they all had major concerns. This will require extensive new experiments to address their comments if you choose to send in a revised manuscript.

One major underlying concern of the reviewers, expressed most strongly by Reviewers 1 and 3, was that there is no clear evidence that the isolates were truly “environmental”. For example, it can not be ruled out that they came from an animal host. This led to questions as to whether these new strains were really outliers relative to the range of known C. albicans clinical isolates, which negatively impacted the significance. I realize this is a difficult question to address, but new experiments will be required to address this point to gain stronger support from the reviewers.

Another weakness pointed out by the reviewers is that the manuscript was generally descriptive. Phenotypic differences were observed, but the underlying mechanisms were not identified. This was also a weakness for the evolution studies. These concerns will require significant new experiments to address.

The reviewers also had major concerns about the design and execution of the experiments. For example, all three reviewers had questions about the evolution studies. Reviewer 1 had questions about the several experiments, including the macrophage studies and the statistical analysis of the data. Reviewer 3 pointed out the lack of standard conditions for different experiments, which made it difficult to compare results. Addressing these concerns will also require new experiments.

**Journal Requirements:**

1) Please provide an Author Summary. This should appear in your manuscript between the Abstract (if applicable) and the Introduction, and should be 150-200 words long. The aim should be to make your findings accessible to a wide audience that includes both scientists and non-scientists. Sample summaries can be found on our website under Submission Guidelines:

https://journals.plos.org/plospathogens/s/submission-guidelines#loc-parts-of-a-submission

2) We noticed that you used the phrase 'data not shown' in the manuscript. We do not allow these references, as the PLOS data access policy requires that all data be either published with the manuscript or made available in a publicly accessible database. Please amend the supplementary material to include the referenced data or remove the references.

- ® on pages: 19, and 21

- TM on page: 21.

5) We have noticed that you have uploaded Supporting Information files, but you have not included a list of legends. Please add a full list of legends for your Supporting Information files after the references list.

Potential Copyright Issues:

i) Figure 5A. Please confirm whether you drew the images / clip-art within the figure panels by hand. If you did not draw the images, please provide (a) a link to the source of the images or icons and their license / terms of use; or (b) written permission from the copyright holder to publish the images or icons under our CC BY 4.0 license. Alternatively, you may replace the images with open source alternatives. See these open source resources you may use to replace images / clip-art:

7) Please amend your detailed Financial Disclosure statement. This is published with the article. It must therefore be completed in full sentences and contain the exact wording you wish to be published.

2) If any authors received a salary from any of your funders, please state which authors and which funders.

8) Please ensure that the funders and grant numbers match between the Financial Disclosure field and the Funding Information tab in your submission form. Note that the funders must be provided in the same order in both places as well.

**Reviewers' Comments:**

Reviewer's Responses to Questions

**Part I - Summary**

Reviewer #1: This manuscript compares three C. albicans isolates from oak trees in the UK to the lab type strain SC5314, a human bloodstream isolate, to assess phenotypic differences that may be relevant to environmental to human adaptation and infection. As expected these strains have substantial genomic and phenotypic differences relative to the lab strain. One of these strains is less fit in standard sugar-replete laboratory conditions and repeated passage in sugar-rich media resulted in one lineage with more pathogenic-like traits. This was associated with greater hyphal growth and cell damage.

In general this is an interesting study using rare environmental isolates and this paper assumes that the site of isolation makes them fundamentally different. This is not what the data shows: these strains are well within the phenotypic diversity that has long been known for C. albicans. Differences in host cell interactions and propensity for hyphal morphogenesis are well established amongst clinical and commensal strains; it does not appear that these oak tree-associated strains are unusual in their variation. If they were considered more in the context of the known phenotypic diversity, it would be a useful complement to the literature. One of the strains appears to represent a new genomic clade, but this is not a focus of this manuscript.

There is an assumption that these environmental isolates, which are quite rare relative to human isolates, represent a latent environmental reservoir of C. albicans that seeds human colonization. This is the primary basis for the significance of this manuscript, but there is simply no evidence to suggest this is true. Humans are not the only warm-blooded animals capable of harboring C. albicans and it is more likely that these isolates are deposited from animal contamination rather than populations of proliferating fungi on trees. The genomic diversity of these isolates from the same type of tree in the same national park is more consistent with bird droppings (for example) than some reservoir with which humans come in contact. The authors are right that there is much unknown about the acquisition and stability of Candida colonization in humans, but the evidence points to person-to-person transmission (e.g., mother to child) rather than environmental contacts and while some speculation otherwise is healthy, this goes a bit far.

There are several smaller concerns that should be addressed, as outlined below.

There is some lack of context in the comparisons between SC5314 and the oak isolates. Figure 2 uses several other human isolates, and this is useful, as it shows that SC5314 is the outlier in some respects and Oak 1 and Oak 3 are outliers in others. But most experiments don’t include other human isolates. Are the oak isolates unusually nutrient restricted or do they have more divergent ECE1 sequences? This would be useful information to put these strains on a more global phenotypic map.

In Figure 1A-B, the growth rates relative to SC5314 are presented as a heat map. The legend says the color coding is based on the p-value, but the scale is more consistent with it being based on a ratio of growth rates, and this matches the text as well. This should be clarified.

When there is a significant difference in hyphal morphology, as there is within macrophages (Fig. 4), CFU plating is not a reliable means to quantitate cell number. One cannot conclude from this data that Oak 1 survives better in macrophages, it just forms individual yeast rather than multinucleated hyphae. With this sort of morphology difference, a metabolic assay (XTT), enumeration of nuclei, or qPCR for fungal DNA would be more accurate choices.

In the same experiments, assaying macrophage survival at 24 hours is problematic given the literature that shows that these cells die from glucose depletion well before that (this is also apparent in Fig. S7, where the background macrophage damage rate is ~60%). This should really be done at shorter (~6h) timepoints.

The “evolution” experiment identifies no genomic changes that are worth elaborating upon, but a considerable transcriptomic change. How do the authors imagine this happening? Epigenetic? Some discussion is warranted.

The statistical analysis is incomplete in places. Data for the Oak strains are compared to SC5314, but in the experiments with the Evo strain it should be compared to the parent as well.

Fig. S8 is important. There are really no differences in metrics of immune activation amongst these strains. This argues that the differences in outcomes (invasion, hyphal growth, damage, etc.) are fungal-intrinsic, i.e., due to strain variation rather than differential recognition by the macrophages. This should be emphasized more in the main text in the discussion around Fig. 4.

Reviewer #2: Lange et al. have analyzed three environmental isolates of *Candida albicans* originally isolated from oak trees in the United Kingdom. They determined the differences between these isolates and the standard C. albicans isolate SC5314 for many properties including host damage, hyphal formation, cellular invasion, and production of candidalysin, a C. albicans protein that causes cell damage. One strain behaved similarly to clinical isolates while the other two were less virulent. The more virulent strain had reduced susceptibility to the antifungal amphotericin B, while all three strains had a reduced susceptibility to fluconazole. One less-virulent strain was evolved over months on galactose, a sugar that is part of the human diet. The evolved strain became more virulent, more metabolically flexible, more able to damage epithelial cells, and more resistant to Amphotericin B. This suggests that virulence and antifungal resistance can be evolved in an environmental strain, and that adaptation to sugar rich diets can affect virulence and drug resistance.

The manuscript is well written. The data for the most part of well-presented and the interpretations are valid without overinterpretation. The conclusions are justified.

Reviewer #3: The authors characterized environmental *Candida albicans* isolates and show that evolution in presence of dietary sugars of a non virulent strain results in a more virulent strain. They also show that such isolates may already be resistant to antifungal drugs.

The authors used various approaches to make their point that non-virulent strains may become virulent under environmental pressure conditions.

It is mainly a descriptive manuscript were the molecular basis of the different phenotyeps are not really explored.

The paragraph on the possible role of Ece1 and its expression level is hypothetical and the possible mechanisms (TATA box location or presence) have not been explored. Also the secretion of the peptides does not reaslly explain the higher damage potential of oak2. This whole part is rather speculative.

**Part II – Major Issues: Key Experiments Required for Acceptance**

Reviewer #1: (No Response)

Reviewer #2: 1. Evolving a strain over months requires precise sterile techniques. Is it clear that the Evo strain and the Oak2 strain are the same strain with minor changes. Can you rule out a different strain contaminating the culture. It is important to state that.

2. Fig 5A is very confusing. The left and center flasks are clear. What is the third flask and what does it represent? What is meant by adaptation and selection for the right flask? Isn’t that going on in the center flask? The figure says samples are taken from the center and right flasks every 3 days. How is the right flask different from the center flask. This needs to be explained better in the text and in the figure legend.

3. The text needs to more closely reflect the growth conditions in the Figure 1,

4. As this is PLOS Pathogens, it would seem necessary to show the behavior for the strains in a OPC or systemic animal model. Maybe competition in either model rather than showing each strain separately. At least OAK2 and EVO separately or in competition.

Reviewer #3: The oak 2 strain shows many characteristics of a virulent strain. How can it be excluded that this is a ‘contaminant’ strain. Can such strains be reisolated from that oak tree?

A phylogenetic tree that places the three oak strains among sequenced isolates from the different Candida clades should be done.

The authors write that growth on glucose is very different between WT and the oak strains, but on YPD (supplementary figure 1), this is not clear. What may be the reason.

It is also unclear why for AMB and FLU different conditions are used (30 ° for 2 days versus 37 °C for 6 days). I think that the FLU data should also be in figure 1 with data shown at the same temperature and same time points.

More importantly here, Figure 1 has been performed at 30 °C but data should be provided at 37 ° C in order to translate better to the human condition.

What is the molecular mechanism of the higher AMB resistance of the Oak 2 strain? This should be the result of a mutation. What does the whole genome sequnce of this strain says?

The oak strains grow less well in glucose medium. For the damage experiment on the epithelial cells, should more oak mutant cells be used compared to the WT. Over a 24 h period, this may result in a growth difference.

For the evolution experiments in the text says YNB+Gal but in legend is SD+Gal. The authors evolved in galactose to mimick the sugar that may not be taken up in small intestine. However, there is also Candida in the small intestine. It would be interesting to also evolve the strain in glucose containing medium.

Apparentely some evolved strains (being able to grow well in galactose) were not used further on, but only one ‘evo’ strain was used. What does the whole genome sequence tells regarding the differences with the parental strain? The authors did this, but did not really find any interesting mutation and only one change in a coding sequence. This raises the question whether the evolved strains were passaged multiple times under control conditions to rule out that it were just adapted strains but not mutated strains?

It is strange that a strain that did not grow on glucose nor galactose, was evolved in galactose and then starts to grow very well on glucose and still very poor on galactose. What is the molecular mechanism?

**Part III – Minor Issues: Editorial and Data Presentation Modifications**

Reviewer #1: (No Response)

Reviewer #2: 1. Fig 2F – why is Oak 3 not included in the analysis as it is included in all other panels4.

2. Line 95 – says that there was reduced growth in all texted alternative nitrogen sources? What about arginine where there is no reduced growth?

3. Line 97 – says that Oak 2 grew similarly to SC5314 on the texted nitrogen sources – what about Tryptophan and some concentrations of ammonium sulfate.

4. Line 201 – There is no Figure 5E.

Reviewer #3: Why is the interaction with the intestinal epithelial cells for 48 h and the oral ones for 24hrs.?

In figure 2F, is it correct that survival in whole blood is only 2-4 % after 15 min? Why is strain oak3 not presented here?

Strain Oak3 is also missing in figure 3D.

Panel C is not indicated in the legend of figure 4. In 4C, I do not see the monocytes in the panel of SC5314. Are they already destroyed by the Candida? Oak1 and Oak3 are not able to escape?

Would it not have been interesting to take the most virulent Oak strain in the evolution study. The effect might even be bigger

Is the MIC for Flu or Amb changed in the evolved strain?

In line 275 it is indicated that ECE1 expression is not changed, but then 10 lines further major conclusion is ECE1 expression is increased in evolved strain after 3 hours. Was it known that the transcript of ECE1 is not stable?

It is quite surprising to see the major differences between in vitro data and the on chip data (which are actually also in vitro).

How far away is the oak tree from the nearest field where agricultural azoles may have been used?

PLOS authors have the option to publish the peer review history of their article (what does this mean? ). If published, this will include your full peer review and any attached files.

**Do you want your identity to be public for this peer review?** For information about this choice, including consent withdrawal, please see our Privacy Policy .

Reviewer #1: No

Reviewer #2: No

Reviewer #3: No

**Figure resubmission:**

**Reproducibility:**



---

## [Decision Letter · Decision Letter 1]

17 Sep 2025

Dear Prof. Hube,

We are pleased to inform you that your manuscript '“Pour Some Sugar on Me” – Environmental *Candida albicans* Isolates and the Evolution of Increased Pathogenicity and Antifungal Resistance through Sugar Adaptation' has been provisionally accepted for publication in PLOS Pathogens.  All three reviewers thought that the new experiments and revisions to the text addressed their concerns.

Best regards,

James B. Konopka

Academic Editor

PLOS Pathogens

Debra Bessen

Section Editor

PLOS Pathogens

Sumita Bhaduri-McIntosh

Editor-in-Chief

PLOS Pathogens

orcid.org/0000-0003-2946-9497

Michael Malim

Editor-in-Chief

PLOS Pathogens

orcid.org/0000-0002-7699-2064

All three reviewers agreed that their scientific issues were addressed in the revised manuscript.

Reviewer #1:

Reviewer #2:

Reviewer #3:

Reviewer Comments (if any, and for reference):

Reviewer's Responses to Questions

**Part I - Summary**

Reviewer #1: The authors have addressed my prior concerns thoughtfully and I have no further reservations regarding this manuscript.

Reviewer #2: See previous summary

Reviewer #3: (No Response)

**Part II – Major Issues: Key Experiments Required for Acceptance**

Reviewer #1: (No Response)

Reviewer #2: See previous issues - all have been addressed

Reviewer #3: The authors nicely addressed all the questions that I had and performed additional experiments as requested

**Part III – Minor Issues: Editorial and Data Presentation Modifications**

Reviewer #1: (No Response)

Reviewer #2: See previous issues - all have been addressed

Reviewer #3: (No Response)

PLOS authors have the option to publish the peer review history of their article (what does this mean? ). If published, this will include your full peer review and any attached files.

**Do you want your identity to be public for this peer review?** For information about this choice, including consent withdrawal, please see our Privacy Policy .

Reviewer #1: No

Reviewer #2: No

Reviewer #3: **Yes: ** Patrick Van Dijck

---

## [Editor Report · Acceptance letter]

Dear Prof. Hube,

We are delighted to inform you that your manuscript, "“Pour Some Sugar on Me” – Environmental *Candida albicans* Isolates and the Evolution of Increased Pathogenicity and Antifungal Resistance through Sugar Adaptation," has been formally accepted for publication in PLOS Pathogens.

Best regards,

Sumita Bhaduri-McIntosh

Editor-in-Chief

PLOS Pathogens

orcid.org/0000-0003-2946-9497

Michael Malim

Editor-in-Chief

PLOS Pathogens

orcid.org/0000-0002-7699-2064